# Assembly of CRISPR ribonucleoproteins with biotinylated oligonucleotides via an RNA aptamer for precise gene editing

Jared Carlson-Stevermer[1,2], Amr A. Abdeen[1], Lucille Kohlenberg[1], Madelyn Goedland[1,2], Kaivalya Molugu[1], Meng Lou[1] & Krishanu Saha [1,2]

Writing specific DNA sequences into the human genome is challenging with non-viral gene-editing reagents, since most of the edited sequences contain various imprecise insertions or deletions. We developed a modular RNA aptamer-streptavidin strategy, termed S1mplex, to complex CRISPR-Cas9 ribonucleoproteins with a nucleic acid donor template, as well as other biotinylated molecules such as quantum dots. In human cells, tailored S1mplexes increase the ratio of precisely edited to imprecisely edited alleles up to 18-fold higher than standard gene-editing methods, and enrich cell populations containing multiplexed precise edits up to 42-fold. These advances with versatile, preassembled reagents could greatly reduce the time and cost of in vitro or ex vivo gene-editing applications in precision medicine and drug discovery and aid in the development of increased and serial dosing regimens for somatic gene editing in vivo.

[1] Wisconsin Institute for Discovery, University of Wisconsin-Madison, Madison, WI, USA. [2] Department of Biomedical Engineering, University of Wisconsin-Madison, Madison, WI, USA. Correspondence and requests for materials should be addressed to K.S. (email: ksaha@wisc.edu)

Precise editing of DNA sequences in the human genome correct mutations[1–3] or introduce novel genetic functionality[4] for many biomedical purposes. Specifically, non-viral delivery of preassembled CRISPR ribonucleoproteins (RNPs) is currently being developed for somatic gene-editing applications[1–3, 5]. RNPs combining *Streptococcus pyogenes* Cas9 nuclease

**Fig. 1** Design of S1mplexes with *Sp*Cas9. **a** Schematic showing preassembled ssODN-S1mplexes that are complexes of *Sp*Cas9 protein, sgRNA with a S1m aptamer, streptavidin, and a single-stranded oligodeoxynucleotide (ssODN) donor template. S1m-sgRNAs add an RNA aptamer at a stem loop of the sgRNA that is capable of binding streptavidin protein. A biotin-ssODN is then added to this tertiary complex. ssODN-S1mplex particles are designed to promote homology directed repair (HDR). **b** Sequence and secondary structure of each S1m-sgRNA. Protospacer designates the region of the sgRNA that defines the sequence to target in the human genome. S1m loop (red) binds streptavidin

(SpCas9, a high-affinity nuclease isolated from a type II CRISPR–associated system) and a single-guide RNA (sgRNA) generate on-target DNA double-strand breaks (DSBs) with little to no off-target DNA cleavage[5, 6]. This break is predominately repaired through one of two major DNA repair pathways: error prone non-homologous end-joining (NHEJ) or precise homology directed repair (HDR), in which a template is used for precise gene editing. Co-delivery of a nucleic acid donor template with

the SpCas9 RNP (SpCas9 + sgRNA) is capable of producing precise edits at target loci through HDR of the DSB. However, variable delivery of the CRISPR system along with the donor templates generates a spectrum of edits, where a majority of cells include imprecise insertions and deletions (indels) of DNA bases from NHEJ or microhomology mediated end-joining of the DSB[7, 8]. Even when precise HDR of the DSB occurs on one allele, there is a chance that both alleles are not identically edited, resulting in

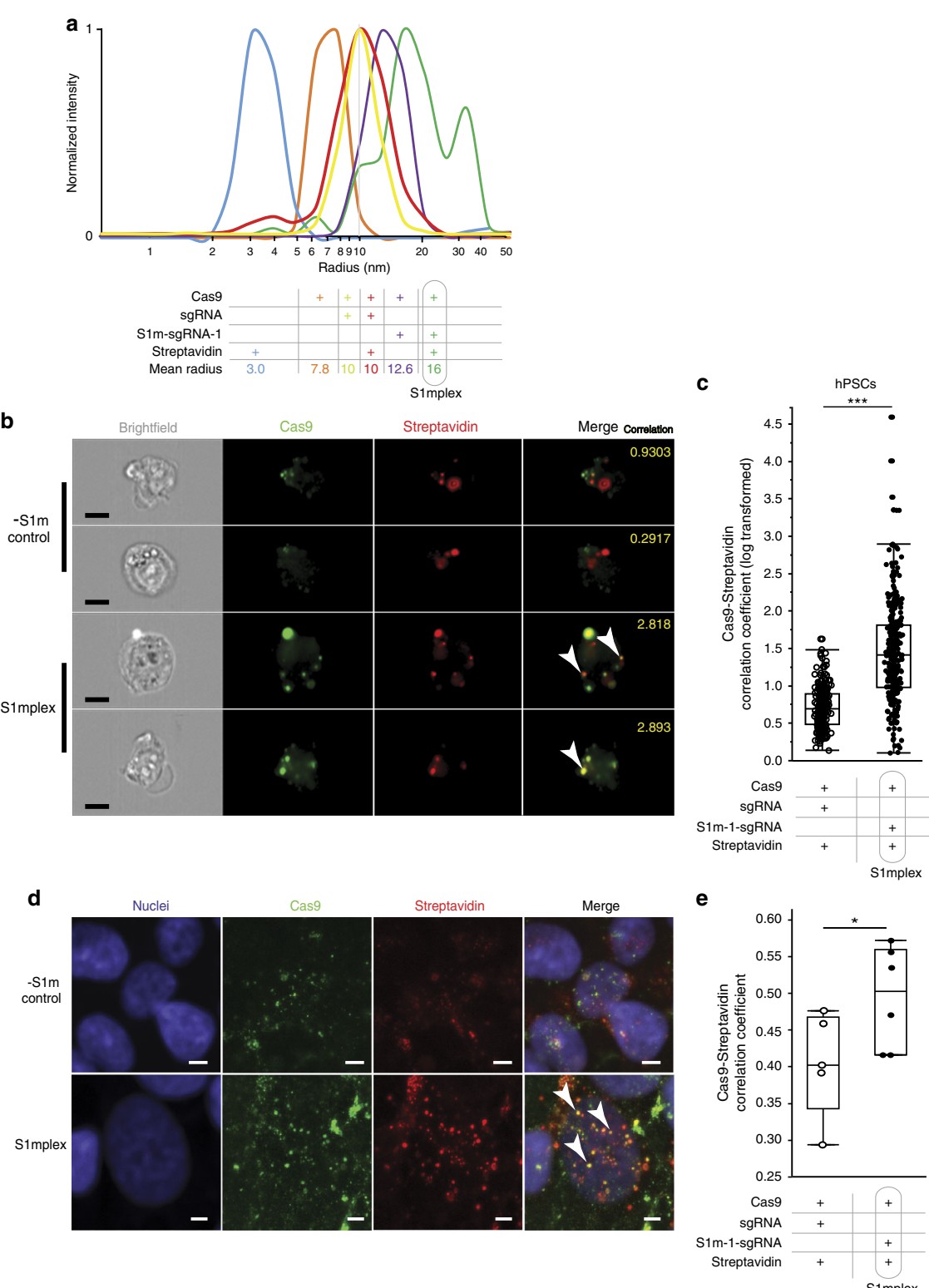

imprecise edits on the other allele[9, 10]. Faithful writing of DNA, or scarless gene editing[11], within human cells at high efficiency remains an outstanding challenge.

Strategies to promote HDR include addition of small molecules to block NHEJ and restrict SpCas9 activity to particular phases of the cell cycle[12, 13], but variability and toxicity has been observed across human cell lines when applying small molecules to promote HDR[14, 15]. Also, selection strategies through viral integration and excision of drug[16] or cell-surface[1] selection cassettes, flow cytometry for co-expressed fluorescent proteins[10, 17, 18], or through transient drug selection (Steyer B.G. et al., submitted) can assist in the isolation of cells with one or two precisely edited alleles[7]. For all of these strategies, imprecise editing through NHEJ typically outnumbers precise HDR outcomes[2, 6]. None of the current strategies precisely control the delivery of the RNP with the donor template, and many resort to "flooding" the cell with high Cas9 expression and donor template. We reasoned that some of the noise in gene-editing outcomes could be reduced by preassembling RNPs with the donor template or other moieties that enable the isolation of precisely edited cells.

The S1mplex tool described here exploits high-affinity interactions between a short RNA aptamer and streptavidin to promote more faithful writing of the human genome. These RNP-containing complexes can be assembled outside the cell to a desired stoichiometry and delivered as an all-in-one gene-editing nanoparticle together with a donor nucleic acid template. In addition, the complexes can be easily decorated with additional moieties such as fluorophores or Qdots to enrich for edited cells. Use of these particles with a biotinylated ssODN reduced heterogeneity in delivery among RNPs and nucleic acids within human cells and enriches the ratio of precisely edited to imprecisely edited alleles up to 18-fold higher than standard RNP methods, approaching a ratio of four precise edits to every one imprecise edit. Further functionalization with a unique fluorophore enables multiplexed editing and enrichment of precisely edited populations through cell sorting. Taken together, advances with the S1mplex tool generates chemically-defined reagents to promote precise editing of the human genome.

## Results

**Design of chimeric sgRNA to bind streptavidin**. We devised a strategy inspired by CRISPR display[19] that leverages structural studies of the RNP to identify locations in the sgRNA sequence where RNA aptamers could be tolerated (Fig. 1a). Three sgRNAs with a modification either in a stem loop of the sgRNA or at the 3′ end were designed (Fig. 1b), as these locations have previously been shown to tolerate additions with a minimal loss in Cas9-binding activity[20]. Separately, at each location, we added perfectly complementary 10 nucleotide block previously shown to aid

aptamer addition to sgRNAs[19] and a 60 nucleotide S1m aptamer[21], which has a strong non-covalent interaction with streptavidin. The added sequence extends the sgRNA stem loop and contains two distinct bulges used for binding. We termed these new sgRNAs S1m-sgRNA-1, S1m-sgRNA-2, and S1m-sgRNA-3 in reference to their position in the sgRNA from 5′ to 3′ (Fig. 1b).

We confirmed that S1m-sgRNAs can be made rapidly in vitro via one-pot transcription[22] and are larger than standard sgRNAs when analyzed by agarose gel electrophoresis (Supplementary Fig. 1a). Next, we verified the ability of S1m-sgRNAs to complex with streptavidin in vitro by combining a constant amount of S1m-sgRNA with increasing amounts of streptavidin (0.1, 1, and 10 molar equivalents) via an electrophoretic mobility shift assay (EMSA). The electrophoretic front of each S1m-sgRNA slowed as streptavidin levels increased (Supplementary Fig. 1b). However, S1m-sgRNA-3 demonstrated the least shift suggesting that the secondary structure necessary for binding may be partially disrupted at this location. At all levels of streptavidin, the electrophoretic front slowed, demonstrating that S1m-sgRNA-1 and S1m-sgRNA-2 interact with streptavidin. In contrast, when the same amount of standard (non-S1m) sgRNA was run with 10 molar equivalents of streptavidin, the electrophoretic front remained constant (Supplementary Fig. 1b).

To demonstrate the ability of S1m-sgRNA-1 to complex with streptavidin and Cas9 protein simultaneously, we performed dynamic light scattering (DLS). When streptavidin and Cas9 were mixed in solution, two peaks were distinct at 3.0 and 7.8 nm (Fig. 2a), both of which match closely the radii previously reported for each protein[23, 24]. We next formed Cas9 RNPs with excess standard sgRNAs and observed that the species formed were larger than Cas9 alone and did not increase in radius with the addition of streptavidin. Excess sgRNA was not detected by DLS and was included in the DLS studies to ensure all key components were able to assemble together (Supplementary Fig. 1c). Additionally, these samples had a discernable peak corresponding to the presence of streptavidin alone. RNPs containing S1m-sgRNA-1s and Cas9 protein increased in radius by a larger amount than RNPs containing standard sgRNAs and Cas9 protein, likely due to the increased length of S1m-sgRNA-1. When streptavidin was added to S1m-sgRNA-1 RNPs, the average radius of the complex was increased by ~3 nm (green arrowhead), the radius of streptavidin protein. These tertiary complexes of Cas9, S1m-sgRNA-1, and streptavidin are termed "S1mplexes." The second, larger peak in the S1mplex DLS trace (green asterisk) is attributed to the tetrameric nature of streptavidin that can harbor up to four RNPs.

While assembly of S1mplexes in vitro is important, the maintenance of complexes post delivery is imperative to gene-editing function. To demonstrate this capability, we delivered Cas9 protein and streptavidin in combination with either sgRNAs

**Fig. 2** Measuring and visualizing S1mplexes in vitro and within cells. **a** Dynamic light scattering of ssODN-S1mplex particle assembly. Cas9 (orange) and streptavidin (blue) proteins fail to assemble when in solution together and have a hydrodynamic radius consistent with published data. The addition of sgRNA to Cas9 protein increases the radius of the RNP particle to 10 nm (yellow). This radius does not change with the addition of streptavidin (red) and free streptavidin can be detected (red arrowhead). When S1m-sgRNAs are added to Cas9 (purple), the radius is increased by a larger amount than sgRNAs, potentially due to the larger size of the S1m-sgRNA-1. When streptavidin is added to this complex (green), a shift in size of about 3 nm occurs (green arrowhead), the size of streptavidin. A second peak at 35 nm (green asterisk) may be associated with multiple Cas9-S1m-sgRNA complexes connected to a single streptavidin. See also Supplementary Fig. 1c. **b** Two representative single cell multispectral flow cytometric images of S1m-sgRNA-1 and sgRNA-transfected cells with Cas9 immunohistochemistry and fluorescent streptavidin (scale bar: 10 μm). Arrowheads indicate presence of overlapping colors. Numbers in yellow are measured log Pearson correlation coefficient as determined by IDEAS software (see also Supplementary Fig. 2). **c** Correlation coefficient of Cas9 immunocytochemistry fluorescent signal and streptavidin fluorescence, as measured by multispectral image cytometry within hPSCs. Use of S1m-sgRNA-1 significantly increased the correlation between the two signals (***$p < 10^{-5}$, Student's two-tailed t test). **d** Representative confocal images of S1m-sgRNA-1 and sgRNA-transfected cells with Cas9 immunohistochemistry and fluorescent streptavidin (scale bar: 5 μm). Arrowheads indicate presence of overlapping colors. **e** Correlation coefficient of Cas9 immunocytochemistry and streptavidin fluorescence inside the nuclei of transfected cells. Introduction of S1m-sgRNAs significantly increased the correlation between the two molecules (*$p < 0.05$, Student's two-tailed t test)

or S1m-sgRNAs into human pluripotent stem cells (hPSCs) via nucleofection and conducted immunohistochemistry for the two protein components. Multispectral imaging flow cytometric analysis of single fixed cells confirmed the co-localization of the two protein components within hPSCs (Fig. 2b, Supplementary Fig. 2). Significantly higher correlation in the fluorescent signals from the two protein components were seen when S1m-sgRNA-1 was included ($p < 10^{-5}$, Student's two-tailed $t$ test, Fig. 2c). To gain further subcellular resolution of these components after

S1mplex delivery, images obtained using confocal microscopy on fixed, intact hPSC cultures were analyzed using CellProfiler[25] for overlap between the two components within the nuclei. At 24 h after delivery, the correlation between the fluorescent signals arising from Cas9 and streptavidin within the nucleus was significantly higher when using S1m-sgRNAs than sgRNAs ($p < 0.05$, Student's two-tailed $t$ test, Fig. 2d, e). Together, these results indicate that complexes between Cas9 and streptavidin are preserved specifically through the S1m aptamer during transfection and subsequent subcellular trafficking such as nuclear transport.

Next, we examined the ability of S1m-sgRNAs to edit genes within human cells. We created a human embryonic kidney (HEK) cell line that constitutively expressed blue fluorescent protein (BFP) from an integrated transgene[26]. DSBs produced by sgRNAs that target the fluorophore in combination with Cas9 expressed from a transfected plasmid are repaired predominantly through NHEJ, with indel formation at the DSB. NHEJ-mediated gene edits are expected to result in a loss of BFP fluorescence within this HEK line. After delivery of S1m-sgRNAs and a plasmid encoding Cas9 to this HEK line, BFP expression was analyzed via flow cytometry. All S1m-sgRNAs (1, 2, and 3) created indels at approximately half the frequency of standard sgRNAs (Supplementary Fig. 3a). While the approximately twofold decrease in generating indel edits is significant, such decreases in indel formation have been linked to a concomitant decrease in off-target effects[27].

**Assembly of DNA repair template to RNP.** We subsequently searched for a method to combine a donor DNA template with S1mplexes and form a quaternary complex in order to promote precise editing through HDR. Given the strong interaction between streptavidin and biotin ($K_D = 10^{-15}$ M), we selected biotinylated single-stranded oligodeoxynucleotide (ssODNs) donor templates. All components (S1m-sgRNA, streptavidin, biotin-ssODN) were run individually on a gel and compared side-by-side with standard reagents (sgRNA, ssODN) to establish baseline migration characteristics via an EMSA. The biotin-ssODN ran slightly larger than the standard ssODN, presumably due to the biotin modification (Supplementary Fig. 3b, lanes four and five). Tertiary complexes of S1m-sgRNA-1, streptavidin, and ssODN were formed using 0.5, 1, or 2 molar equivalents of biotin-ssODNs. The primary band displayed a distinct electrophoretic shift from either the sgRNA or ssODN alone, indicating complex formation (Supplementary Fig. 3b, lanes six to nine). To demonstrate that all components combined successfully,

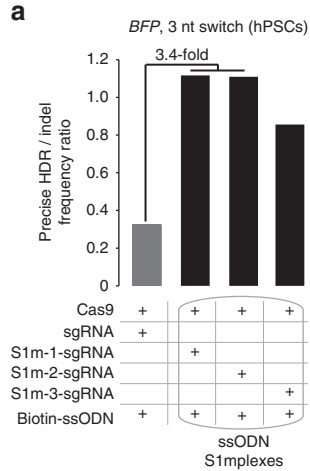

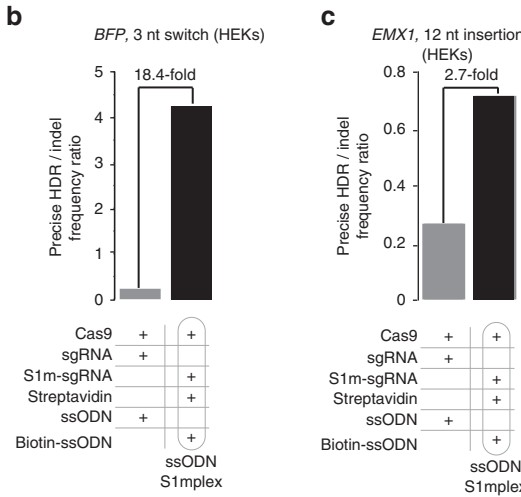

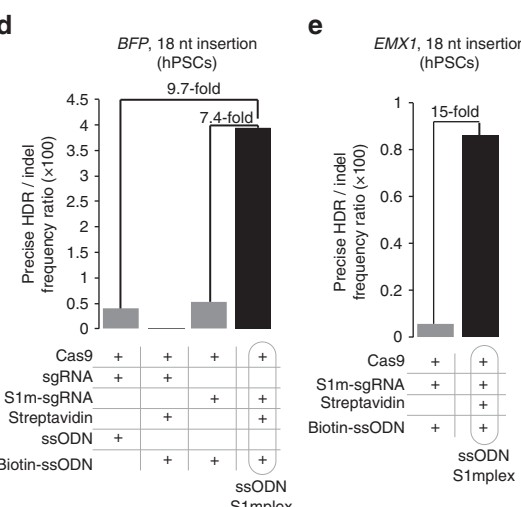

Fig. 3 ssODN-S1mplexes increase ratio of HDR to indels in human cells. **a** Ratio of precise to imprecise editing using S1mplexes formed with different S1m-sgRNA variants in hPSCs. Each S1m-sgRNA increased the ratio of precise to imprecise editing when compared to sgRNAs. S1mplexes with S1m-sgRNA-1, and S1m-sgRNA-2 had the highest ratios of precise editing. **b** Ratio of precise to imprecise editing at *BFP* locus. ssODN-S1mplexes had an 18.4-fold higher ratio than sgRNAs and contained four precise edits to every one indel as analyzed by deep sequencing 8 days post lipofection of HEKs. **c** Ratio of precise to imprecise editing at *EMX1* locus. ssODN-S1mplexes had a 2.7-fold higher ratio than sgRNAs. **d** Ratio of precise insertions to imprecise indels at *BFP* locus in hPSCs as analyzed by deep sequencing. ssODN-S1mplexes had a 9.7-fold increase in comparison to standard sgRNAs and a 7.4-fold increase when compared with untethered ssODNs. **e** Ratio of precise insertions to imprecise indels at *EMX1* locus. Addition of streptavidin to S1mplex resulted in a 15-fold increase in the ratio of precise insertions to imprecise indels. See also Supplementary Table 9

unmodified ssODNs were run in the place of biotin-ssODNs. The unmodified ssODN displayed the expected electrophoretic shift despite the presence of the S1m-streptavidin complex (Supplementary Fig. 3b, lanes 10–12). Finally, standard sgRNA was run with streptavidin and biotin-ssODN. In this condition, the band attributed to S1m-sgRNA-1-streptavidin or S1m-sgRNA-1-streptavidin-ssODN binding was not observed and instead solid bands representing sgRNA and ssODN-streptavidin were present (Supplementary Fig. 3b, lanes 13 and 14). Due to the strong interaction of biotin and streptavidin, we needed to ensure that biotin did not displace S1m-sgRNA-1 already bound to streptavidin when added in solution. To do so, we combined S1m-sgRNA-1s with streptavidin at a 1:1 molar ratio. We then added fourfold molar excess of biotin to occupy every binding site on each streptavidin molecule and incubated the complex for 0, 5, 10, 20, or 30 min. After incubation, gel shift following electrophoresis was not different from bound S1m-sgRNA:streptavidin combinations suggesting that biotin did not interfere with the S1m-streptavidin interaction at the concentrations used in this study (Supplementary Fig. 3c).

**Increased HDR to indel ratios in human cells**. We tested the ability of all three ssODN-S1mplexes to induce HDR in a hPSC line containing a BFP-expressing transgene that can be switched to express GFP through a 3 nucleotide switch (Supplementary Fig. 4a)[26]. S1mplexes with biotin-ssODNs (ssODN-S1mplexes) were assembled using one of the three S1m-sgRNAs and compared to standard sgRNAs and ssODN combinations. After delivery of ssODN-S1mplexes and subsequent deep sequencing of genomic DNA, we found that all three ssODN-S1mplexes had a higher ratio of HDR:indel editing than standard RNPs. ssODN-S1mplexes with S1m-sgRNA-1 and S1m-sgRNA-2 induced similar ratios of HDR:indel editing while ssODN-S1mplexes with S1m-sgRNA-3 had a slightly depressed HDR:indel ratio (Fig. 3a). The decreased HDR:indel ratio found using S1m-sgRNA-3 may have been due to the lower binding affinity of this sgRNA with streptavidin, as seen in the EMSA (Supplementary Fig. 1b). In order to minimize the frequency of indel mutations while maximizing HDR, we decided to use S1m-sgRNA-1 for all remaining experiments and will refer to it henceforth simply as S1m-sgRNA.

With this knowledge, we then evaluated S1mplexes in multiple human cell lines for their ability to generate a variety of precise nucleotide changes. We assembled ssODN-S1mplexes to again switch BFP to GFP. After delivery to HEK cells, deep sequencing revealed that the ssODN-S1mplexes enriched the ratio of precise insertions to imprecise editing 18.4-fold over standard RNPs and approached a ratio of four precise edits to every one indel (Fig. 3b, Supplementary Fig. 4b, and Supplementary Table 9). When the same experiments were conducted in hPSCs, results from flow cytometry assays were consistent with these conclusions from deep sequencing (Supplementary Note 1, Supplementary Fig. 4c–f, and Supplementary Table 9). Additionally, when introducing a 12 nucleotide insertion into the EMX1 locus[28] of HEKs with ssODN-S1mplexes, the ratio of precise insertions to imprecise editing increased 2.7-fold over standard sgRNA RNPs (Fig. 3c, Supplementary Fig. 4g, and Supplementary Table 9). Taken together, this shows that ssODN-S1mplexes are able to shift the balance of editing to enrich for small, precise edits within the genome.

We tested the ability of this strategy to create even larger sequence changes in hPSCs by designing an ssODN that carried a variable 18 nucleotide insertion. We deep-sequenced the cell population after delivery of ssODN-S1mplexes, again targeting the BFP and EMX1 loci. When standard sgRNA RNPs were transfected with streptavidin-ssODN complexes, minimal insertion was seen with a low ratio of precise HDR to imprecise indel alleles (Fig. 3d and Supplementary Table 9). Equivalent precise to imprecise editing was seen when standard sgRNA RNPs and ssODNs were transfected, as well as when S1m-sgRNA RNPs were transfected with biotin-ssODN (without streptavidin) (Fig. 3d and Supplementary Table 9). However, levels of indels were increased in the standard sgRNA RNP with free ssODN condition (Supplementary Fig. 4h and Supplementary Table 9). When the full ssODN-S1mplexes were transfected into hPSCs, HDR insertion levels greatly increased (Supplementary Fig. 4i) as did the ratio of precisely edited to imprecisely edited alleles to 9.7-fold over standard RNP methods (Fig. 3d). At the endogenous EMX1 locus, we delivered the S1m-sgRNA RNPs with biotin-ssODNs either with or without streptavidin. When streptavidin was added to generate the full ssODN-S1mplex, rates of insertion increased 51-fold (Supplementary Fig. 4j), and the ratio of precise to imprecise gene editing increased 15-fold (Fig. 3e). Taken together, each component of the ssODN-S1mplex is necessary to drive higher HDR:indel ratios within human cells.

**Design constraints on the ssODN-S1mplex**. Recent studies have reported that the design of the ssODN has a significant effect on the rate of HDR[15, 26]. Accordingly, we explored various ssODN designs with ssODN-S1mplexes. Designs were limited to a 100 nucleotide length due to ease of ssODN synthesis, but varied as follows: asymmetrical around the cut site, extending 30 nt upstream and 67 nt downstream or vice-versa, either identical to the sequence containing the PAM or the reverse complement (non-PAM), and biotinylated on either the 5′ or 3′ end of the ssODN (Fig. 4, left). S1mplexes containing each unique ssODN were assembled and transfected separately into BFP-expressing hPSCs. Four days after delivery, genomic DNA from each condition was collected and analyzed using deep sequencing. Under these conditions, $2.8 \pm 2.2\%$ of alleles in all samples were edited via HDR and NHEJ (Fig. 4a, top and Supplementary Table 9). We observed that neither the asymmetry, sidedness, biotin, nor location on the ssODN had a significant effect on the HDR or indel outcomes using ssODN-S1mplexes (Fig. 4a, top and Supplementary Table 9). Precise editing ranged from 2 to 10 times greater than imprecise editing (Fig. 4b, top and Supplementary Table 9).

We next sought to test these ssODN designs at an endogenous GAA locus using a patient-derived hPSC line[29] that contains a pathogenic 1 bp deletion in exon 10 on one allele. We designed sgRNAs that target only the mutant allele as well as ssODNs to correct the mutation to wild type and modify the PAM site. These ssODNs were again asymmetrical, 34 bp upstream and 66 bp downstream from the cut site, complementary to the PAM or non-PAM strand, and biotinylated at either the 5′ or 3′ end of the ssODN (Fig. 4, bottom). At this locus, ssODN-S1mplexes again had higher levels of precise to imprecise editing than RNPs consisting of sgRNAs, with three to eight precise edits occurring for every imprecise edit (Fig. 4b, bottom and Supplementary Table 9). Consistent with the sequencing results at the BFP locus, absolute levels of HDR and NHEJ editing were $2.0 \pm 1.1\%$ (Fig. 4a, b, bottom and Supplementary Table 9). There was still no significant difference between any of the ssODNs tested when complexed to the S1mplex.

**Adding fluorescent cargoes to the RNP**. To facilitate isolation of the precisely-edited cells, we pursued a strategy to label the cells that received the S1mplexes by including additional biotinylated fluorescent cargoes. We preassembled standard streptavidin-conjugated quantum dots (QdotSA, 20 nm diameter) with S1mplexes (QdotSA-S1mplexes, Fig. 5a, top). After transfection

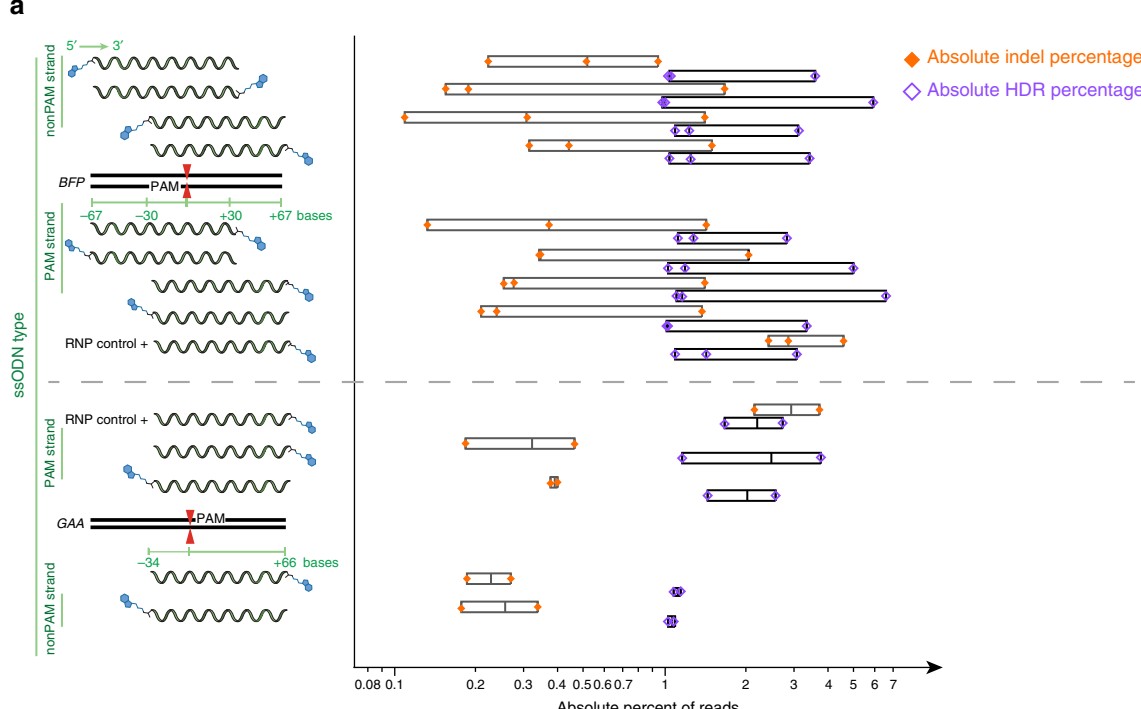

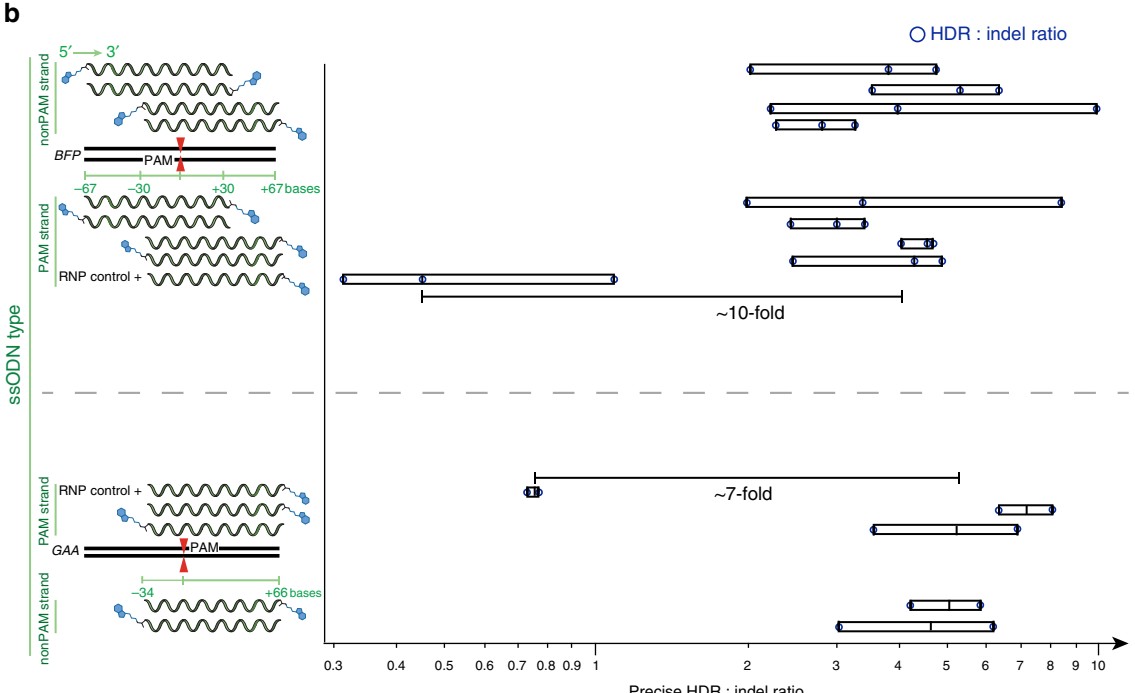

**Fig. 4** ssODN asymmetry, strandedness and tether location minimally affects S1mplex gene editing. Left: ssODN design. Genomic sequence is denoted with black bars. sgRNA target strand and PAM is denoted by "PAM" inside genomic locus, while red triangles are the sgRNA cut site. ssODN length is measured around cut site either upstream (−) or downstream (+) as read by the reading frame. Biotin (blue hexagons) was attached to either the 5' or 3' end of the ssODN. ssODNs were identical in sequence to either the PAM or non-PAM sequence as read in a 5'-3' direction. RNP controls were standard sgRNAs plus corresponding ssODN. **a** Absolute NHEJ (orange diamonds) and HDR percentage (purple diamonds) as a function of total reads at two different loci in hPSCs using different ssODN designs. Each symbol represents a single replicate analyzed by deep sequencing 4 days after nucleofection into hPSCs. HDR levels were generally higher in each replicate than NHEJ levels. See also Supplementary Table 9. **b** Ratio of HDR:indel reads in deep sequencing using each ssODN combined with S1mplexes. Blue circles represent individual biological replicates. With each ssODN, S1mplexes increased the ratio of HDR:indel when compared to sgRNA controls, but no significant trends as to symmetry, sidedness, or biotin location were observed

of QdotSA-S1mplexes into HEKs, a subpopulation of cells contained Qdots within the cytoplasm. High-intensity green fluorescence dots were distributed variably across the transfected cell population, indicating that standard transfection methods likely generate significant heterogeneity in the number of RNPs delivered to each cell. Despite the presence of Qdots in the cytoplasm, very low gene editing was observed upon further culture and analysis within a HEK H2B-mCherry reporter cell line (Fig. 5b, Supplementary Fig. 5), suggesting linkage to Qdots was inhibiting RNP function within inside the cell, perhaps due to the Qdots being too large to allow RNP function within the nucleus. Even when transfecting QdotSA five hours post transfection of S1m-

sgRNA RNPs, an inhibition of gene editing was observed, suggesting QdotSA could complex with and inhibit S1m-sgRNA RNPs within cells (Supplementary Fig. 5).

In the ensuing experiments, we chemically-modified Qdots so that the biotin linkage of the S1mplex to the Qdot was mediated through a enzymatically-cleavable disulfide linker (Qdot-SS-S1mplex, Fig. 5a, bottom). With this cleavable linker, we observed a gain in gene-editing activity (Fig. 5b), while the Qdots remained largely within the cytoplasm (Fig. 5c), suggesting separation and nuclear transport of the RNP. The fluorescence from the Qdot at 24 h post transfection was utilized for fluorescence activated cell sorting (FACS). There was a shift in fluorescence for the whole cell population, indicating uptake of Qdot-S1mplexes in most cells, although to differing extents (Fig. 5d). Sorted cells based on Qdot positive fluorescent signal resulted in gene editing at 3.7-fold higher rates versus cells transfected using standard methods (Fig. 5e).

**Multiplexed gene editing with S1mplexes**. To obtain further control and refine the mutagenic spectrum of S1mplexes, we attached a fluorescent label directly to streptavidin that could be used for identification during flow cytometry. First, we pre-assembled an S1m-sgRNA and biotin-ssODN targeting *BFP* with a streptavidin labeled with a red fluorophore (AlexaFluor-594) (Fig. 6a) and then performed a single cell FACS for the isolation of clones that had high fluorescence after delivery. Upon further cell culture, clones were analyzed by Sanger sequencing for editing at the *BFP* locus. Of the 34 isolated clones in the S1mplex-positive population, eight underwent HDR; eight harbored indels; and, the rest remained unedited (Fig. 6b). In comparison, when using sgRNAs, seven of the 41 isolated clones harbored indels and none were positive for HDR. Cell populations did not contain mosaic gene editing, indicating that defined gene-editing outcomes could be enriched by FACS on the S1mplex fluorescence. Second, using this capability, we tested whether if it was possible to multiplex edits using differently colored S1mplexes. We thus assembled the same ssODN-S1mplex targeting *BFP*, termed red-ssODN-S1mplex, and separately complexed an S1m-sgRNA and biotin-ssODN targeting *EMX1* with a streptavidin labeled with a green fluorophore (AlexaFluor-488), termed green-ssODN-

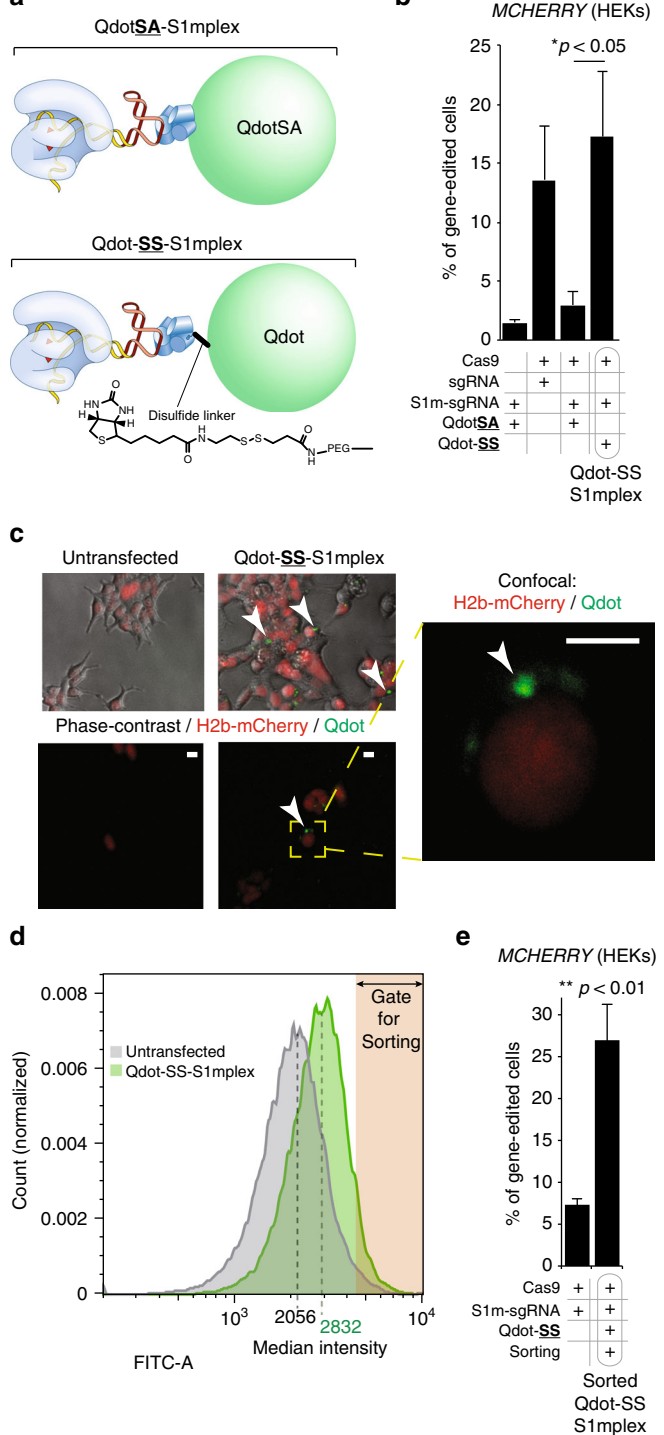

**Fig. 5** Addition of Qdots with degradable linkers to identify S1mplex-positive cells. **a** Schematic of S1mplexes with quantum dot cargoes. Qdots can be complexed with the S1mplex by streptavidin covalently-attached directly to the quantum dot (QdotSA-S1mplex, top), or a enzymatically-cleavable disulfide linker (Qdot-SS-S1mplex, top). The quantum dot has a mean diameter of 20 nm. **b** Gene-editing comparison of different Qdot-S1mplexes. Gene editing of HEK H2B-mCherry reporter cells 5 days post sorting as assayed by flow cytometry. DSBs within *MCHERRY* transgene repaired by NHEJ will generate indels near the fluorophore resulting in a loss of fluorescence. Percent gene editing refers to percent of cells without mCherry fluorescence. QdotSA interferes with RNP activity, while Qdot-SS has equivalent gene-editing activity as the free RNP ($*p < 0.05$, Student's two-tailed $t$ test, error bars $\pm 1$ s.d., $n = 3$ technical replicates). **c** Representative epifluorescence and confocal images of untransfected and Qdot-SS-S1mplex transfected cells 24 h post transfection (scale bar: 10 µm). Arrowheads indicate Qdot fluorescence in the cytoplasm. **d** Increased fluorescence of Qdot-S1mplex (green) allows sorting out of quantum dot positive fractions compared to untransfected cells (grey) 24 h post transfection. **e** Quantum dot conjugation to S1mplex via a cleavable disulfide linker allows fluorescent enrichment of gene-edited human cells ($**p < 0.01$, Student's two-tailed $t$ test, error bars $\pm 1$ s.d., $n = 3$ biological replicates)

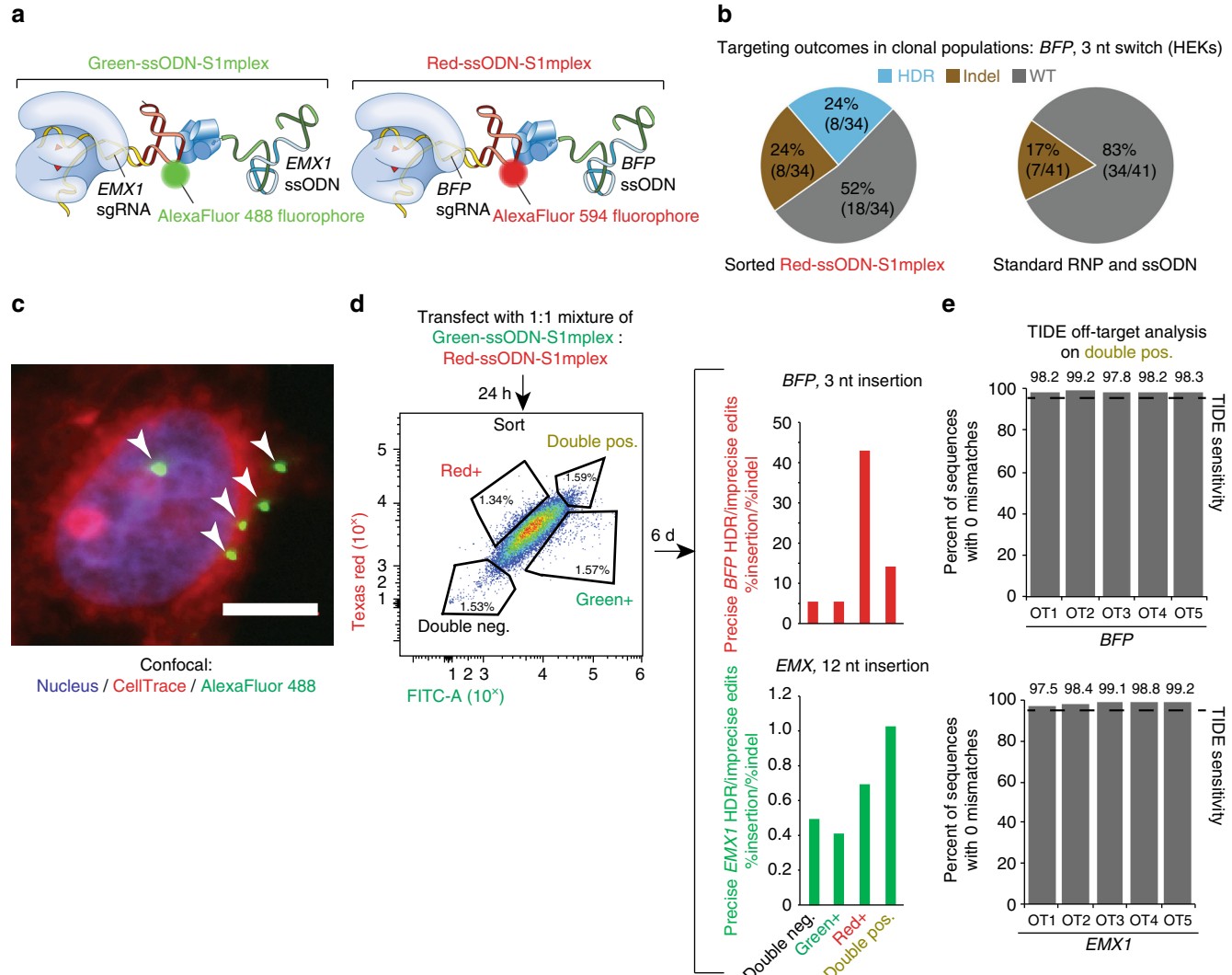

**Fig. 6** Fluorescent S1mplexes to enrich for gene-edited cells. **a** Schematic of simultaneous editing at two loci strategy. HEK cells were transfected simultaneously with two S1mplex particles, labeled with distinct fluorophores. Editing at *BFP* locus was associated with Red-ssODN-S1mplexes (AlexaFluor-594 fluorophore), while editing at *EMX1* locus was associated with Green-ssODN-S1mplexes (AlexaFluor-488 fluorophore). **b** Single cell sorting for enrichment of editing at *BFP* locus. In enriched S1mplex clonal populations, indels (brown) and HDR (blue) events occurred in a 1:1 ratio. In sgRNA clones, all isolated clones either had indel or wild-type genotypes. Genotypes were assayed by Sanger sequencing. No mosaic genotypes were observed. **c** Fluorescent S1mplexes inside the cell using confocal microscopy. Arrows denote Green-S1mplex both inside the nucleus and outside the cell (scale bar: 10 μm). **d** Twenty-four hours post transfection, cells were sorted into populations that were positive for either fluorophore, both or neither. Analysis via deep sequencing was done 6 days post sorting. Top: ratio of precise (perfect sequence match to ssODN) to imprecise editing (indels) in sorted populations. Populations enriched for *BFP* targeted S1mplexes (red positive and double positive) had elevated ratios up to 40 times as many insertions as indels. Bottom: ratio of precise to imprecise editing in sorted populations. Populations enriched for *EMX1* targeted S1mplexes (red positive and double positive) had elevated ratios of precise insertions to indels. See also Supplementary Table 9. **e** Off-target analysis of double-positive populations using TIDE at the top five off-target locations for each sgRNA. No modifications were detected below the TIDE limit of detection (dotted line)

S1mplex (Fig. 6a). The two ssODN-S1mplexes were mixed and transfected simultaneously into HEKs (Fig. 6c).

Twenty-four hours post transfection, we sorted cells using FACS into one of four populations: positive for either fluorophore, both, or neither (Fig. 6d). Only the top 2% of each population was taken, as we observed some association of the fluorescent S1mplex with the cell membrane in addition to robust fluorescent signal within the nucleus of some of the cells (Fig. 6c). One-week post sort, each of the four populations was analyzed for editing via deep sequencing as well as by flow cytometry for *BFP* editing or insert-based PCR for *EMX1*. Deep sequencing revealed that editing at the *EMX1* locus was increased in the presence of green-ssODN-S1mplexes (green positive and double-positive fractions) (Fig. 6d, Supplementary Fig. 6b and Supplementary

Table 9). In these populations, the ratio of precise to imprecise edits increased and approached one and was twofold greater than that of the double-negative fraction (Supplementary Fig. 6b and Supplementary Table 9). Similarly, editing at the *BFP* locus was increased in the red positive and double-positive fractions. As was seen in previous deep-sequencing experiments, the ratio of precise to imprecise edits was elevated in the presence of S1mplexes. With the addition and sorting of fluorescent S1mplexes, the ratio was greater than 10 HDR insertions per indel (Fig. 6d and Supplementary Table 9). Interestingly, the level of indels was highest in the double-negative fraction (Supplementary Fig. 6b and Supplementary Table 9); this may be due to the presence of unlabeled RNPs that did not complex with streptavidin. Results with conventional flow cytometry and PCR

assays followed the same trends, consistent with these conclusions from deep sequencing (Supplementary Note 1, Supplementary Fig. 6c, d, and Supplementary Table 9). We analyzed the top five off-target sites for both the BFP and EMX1 sgRNAs using TIDE[30] in the sorted fractions as well as previous samples used for deep sequencing. None of the sorted populations using ssODN-S1mplexes had modifications above the TIDE limit of detection (Fig. 6e, Supplementary Fig. 7). However, using standard sgRNA RNPs, notable off-target mutagenesis occurred at EMX1 off-target site 2 (Supplementary Fig. 7). Taken together, the assembly of S1mplex particles with a fluorescent tag can be used to create multiple, precise edits with increased efficiency without needing multiple transfections or extended culture.

## Discussion

Together, our results indicate that addition of an S1m aptamer to the sgRNA can generate nanoparticles that shift the balance of precise gene-editing outcomes to outnumber imprecise editing. This shift toward precise gene editing was seen across two different cell lines and three different loci with a variety of insertions and base pair changes. The S1m-streptavidin linkage was strong enough that excess biotin was unable to displace it in vitro. Further, the S1mplex was not observed to dissociate after nucleofection suggesting the complex is stable under a wide variety of conditions. We saw robust S1mplex localization both immediately following and 1 day post transfection. Editing with RNPs has been observed to peak within 24 h of delivery[31]. This supports the idea that the S1mplex is intact within the cell and nucleus. Therefore, S1mplexes may enable HDR by creating a more defined stoichiometry of the RNP to the ssODN within each cell and presumably at the DSB. Each DSB would have at least one local HDR template, which could either suppress NHEJ repair or promote HDR. One group recently reported an approach to localize donor ssODN to the DSB by chemically conjugating the sgRNA with the ssODN and showed increased levels of HDR but no concurrent decrease in indel formation[32]. Further studies characterizing the location of ssODN relative to the DSB within the nucleus may help to clarify the different gene-editing outcomes among differing assembly strategies.

Interestingly, and contrary to recently published studies[15, 26], the asymmetry and sidedness of donor DNA complexed to S1mplexes did not have a significant effect on the ratio of precise editing to imprecise editing. This could be due to the template being readily available for access by HDR mechanisms as opposed to variation based on factors such as cell cycle[12] or cell type. Increased HDR outcomes after Cas9 DNA cleavage have recently been linked to the ataxia-telangiectasia mutated signaling pathway within human zygotes[6, 33] and Fanconi anemia pathways within human hematopoietic progenitor cells[34], leading to the hypothesis that distinct cellular host factors can control the extent of HDR. Given the robust shift toward HDR in both HEK and hPSC lines with the S1mplex, S1mplexes likely interact with common DNA repair machinery found in both HEKs and hPSCs. Further, because the polarity of ssODN biotinylation did not affect the shift toward HDR, differential modifications of the ssODN could be performed selectively at either end to protect from degradation and increase HDR efficiency[35].

Using ssODN-S1mplexes, absolute levels of editing via HDR was 1.6% on average across all cell types and loci, and ranged from 0.052 to 6.6% (Supplementary Tables 8 and 9). Only in one experiment did we see lower levels of HDR using ssODN-S1mplexes when compared to a standard strategy (12 bp insertion into EMX1 within HEKs, Supplementary Fig. 4g and Supplementary Table 9). While less than two previous reports in human cells[26, 36], our absolute editing efficiency is similar to other

reports in pluripotent stem cells in vitro (Supplementary Table 8). Producing HDR at the levels observed in this study while reducing the prevalence of indels has been a key challenge with hPSCs[37]: nevertheless, some studies have been able to target loci that may have important physiological effects[38, 39] and model point mutations within inherited disorder[10] using high-throughput or selection strategies. Also, these levels of HDR can be useful for hotspot mutation modeling[40] and for drug target validation[41] within human cancer cell lines. We note that several reports have suggested that 0.1–15% levels of HDR may have a significant effect in vivo on clinical phenotypes such as Duchenne muscular dystrophy[42], hemophilia B[43], and tyrosinemia type 1[44, 45].

We noticed that HDR levels decreased as the length of insertion increased and the length of homology arms decreased, which has been shown previously[11, 15, 46]. We achieved the highest ratios of precise editing to imprecise editing when generating small changes in an integrated transgene at one allele. This may account for decreased ratios observed at the EMX1 locus where insertions were 20% the length of donor ssODN. Other reports have noted that repair at this allele poses a greater challenge than at the BFP locus, even when attempting small nucleotide changes[26], and that larger homology arms play a significant role in the rate of HDR at this locus[12]. Utilizing larger homology arms for insertion of longer DNA stretches or even transgenes within the genome using S1mplexes has not been explored in this study but warrants further study. There is also the potential at this locus that that repair could be accomplished by a second allele within the genome. HDR using a second endogenous allele for our experiments would be classified as a wild-type deep-sequencing read, so these editing events were not captured in the precise editing HDR frequencies. Importantly, ratios of precise editing to imprecise mutations at the pathogenic GAA allele, which underlies Pompe disease[29], were similar to those found at BFP. This consistency suggests that the variation in HDR:indel ratio observed when compared to large insertions may be due more to the number of nucleotide changes rather than locus-based variation. There may also have been further repair using the healthy GAA allele: however, we were unable to unravel this possibility with certainty in a bulk population. We anticipate that higher ratios of precise to imprecise editing could be generated for single nucleotide changes. 44 750 disease-associated single nucleotide or indel mutations up to 50 nucleotides in length in the ClinVar database[47] can be corrected, in principle, by HDR via donor ssODN templates.

Gene editing in human cells could be controlled by the linkages within the S1mplex. For the Qdot-S1mplexes, a gain of RNP activity occurred only after switching to a labile disulfide bond, presumably because large cargoes such as Qdots (20 nm diameter), assembled with the RNP inhibit Cas9 nuclease activity. The smaller ssODN-S1mplexes without Qdots with mean diameters of 16 nm could generate edits at target loci. The Qdot-S1mplex results demonstrate that the biotin-streptavidin linkage is strong enough to associate biotinylated cargoes with the RNP within human cells, while disulfide bonds, which are labile[48], likely dissociate the S1mplex within cells and release the RNP from the cargo to fully recover activity. Regulating CRISPR gene editing tightly through the release of large cargoes could be explored with other chemistries that generate labile cargoes upon excitation by light or heat[49]. Such strategies could advance targeted therapy to specific areas and cell types within the body and also be applied through a serial dosing regimen, taking advantage of the high precise-to-imprecise editing ratio of S1mplexes.

One study recently reported that biotin-ssODNs could be recruited to RNPs within the cell produced by translation of injected Cas9-avidin messenger RNA (mRNA)[50]. mRNA

strategies still require robust host translational machinery and must avoid an immune response to the delivered foreign mRNA. The Cas9-avidin fusion protein experiments with human cells did not demonstrate lower NHEJ and had a twofold increase in the HDR frequency. These increases are lower than the 3–10 fold increases that we observed at different loci in hPSCs. While the use of fewer components, such as in a Cas9-avidin system, decreases the complexity of editing experiments, our ability to preassemble necessary moieties outside the cell has numerous advantages. The S1mplex strategy offers multiplexing capabilities, such as with the assembly of Qdots to the S1mplex, as well as the potential to purify and isolate only successfully assembled particles. In contrast, in the use of Cas9-avidin mRNA system, ssODN must still be localized to gene-editing machinery within the cell and after translation, which causes significant lag time while still having significant free ssODN within the cells.

The S1mplex strategy provides a straightforward, robust, and modular toolkit to regulate the gene-editing activity of SpCas9 RNPs. RNA modification of the sgRNA with S1m aptamer can be performed readily through short nucleic acid synthesis methods, whereas other strategies that engineer the Cas9 protein can add challenges in protein expression, purification, and stability[50, 51]. Movement of the S1m aptamer around the sgRNA also showed S1mplex formation is viable in each stem loop, suggesting that multiple S1m sequences could be added simultaneously to increase avidity[21] or that other functional aptamers could be added in addition to S1m[19, 52]. Use of RNA modifications could complement and add functionality to already engineered variants of SpCas9 (e.g., high fidelity[53], switchable[54], and optogenetic[55] nucleases). Preassembled, non-viral S1mplexes could also be readily manufactured as off-the-shelf reagents with well-defined critical quality attributes appropriate for clinical use: for example, avidin has previously been tolerated in clinical trials[56], and clinical grade SpCas9 is available from several vendors. We anticipate that the S1mplex strategy will be used as a versatile toolkit to further refine our abilities for precise, scarless gene editing in vitro and potentially in vivo.

## Methods

**Cell culture.** All hPSCs were maintained in E8 medium on Matrigel (WiCell) coated tissue culture polystyrene plates (BD Falcon). Cells were passaged every 3–4 days at a 1:6 ratio using Versene solution (Life Technologies). Parental WA09 line was obtained from WiCell, and WA09-BFP hESCs were generated through lentiviral transduction of BFP dest clone (Addgene #71825) and sorted to ensure clonal populations. After expansion, lines were sorted monthly on a BD FACS Aria to maintain BFP expression levels. Human induced pluripotent stem cell (hiPSC) line Pompe GM04192 was a gift from the T. Kamp and M. Suzuki (UW-Madison) labs.

Human embryonic kidney cells (293T) were obtained from ATCC and maintained between passage 15–60 in growth medium containing DMEM (Life Technologies), 10% by volume FBS (WiCell), 2 mM L-Glutamine (Life Technologies), and 50 U mL$^{-1}$ penicillin–streptomycin (Life Technologies). Cells were passaged 1:40 with Trypsin-EDTA (Life Technologies) onto gelatin-A (Sigma)-coated plates. HEK-H2B-mCherry lines were generated through CRISPR-mediated insertion of a modified AAV-CAGGS-EGFP plasmid (Addgene #22212) at the AAVS1 safe harbor locus using gRNA AAVS1-T2 (Addgene #41818). HEK-BFP lines were generated through lentiviral transduction of BFP dest clone and sorted monthly to maintain BFP expression levels. All cells were maintained at 37 °C and 5% $CO_2$ and verified to be mycoplasma free at least once every month.

**One-pot transcription of S1m-sgRNA variants.** S1m-sgRNAs were synthesized in three steps. Double-stranded DNA blocks encoding the sgRNA scaffold and the S1m aptamer were first created by overlap PCR using Phusion High-Fidelity Polymerase (New England Biolabs) according to manufacturer's protocols. Primer sequences are listed in Supplementary Table 1, and genomic target sequences are listed in Supplementary Table 2. The S1m-sgRNA-1 scaffold was generated under the following thermocycling conditions: 30 cycles of 98 °C for 10 s and 72 °C for 15 s with a final extension period of 72 °C for 10 min. The S1m-sgRNA-2 and S1m-sgRNA-3 scaffolds were generated under the following thermocycling conditions: 10 cycles of 98 °C for 10 s, 61.2 °C for 10 s, and 72 °C for 15 s, followed by 20 cycles of 98 °C for 10 s and 72 °C for 15 s with a final extension

period of 72 °C for 10 min. These scaffolds were combined with a second primer consisting of a truncated T7 promoter, the sgRNA target, and homology to the S1m scaffold and PCR was performed again under the following thermocycling conditions: 98 °C for 30 s followed by 35 cycles of 98 °C for 5 s, 60 °C for 10 s, and 72 °C for 15 s, with a final extension period of 72 °C for 10 min. S1m PCR products were then incubated overnight at 37 °C in a HiScribe T7 IVT reaction (New England Biolabs) according to manufacturer's protocol. The resulting RNA was purified using MEGAclear transcription clean-up Kit (Thermo Fisher) and quantified on a Nanodrop2000.

**S1m-sgRNA RNP formation.** NLS-Cas9-NLS protein (Aldevron, Madison, WI) was combined with S1m-sgRNAs at a 1:1.2 molar ratio and allowed to complex for 5 min with gentle mixing. To this complex, streptavidin (Life Technologies) was added at an equimolar ratio to Cas9 and the mixture was allowed to complex for an additional 5 min. Finally, a 1.2 molar equivalent of biotin-ssODNs (Integrated DNA Technologies) to Cas9 was added to the tertiary complex and subsequently vortexed at low speed. This final mixture was then allowed to combine for 10 min to ensure complex formation. ssODN sequences are listed in Supplementary Table 4.

**S1m-sgRNA and streptavidin-binding gel shift assays.** S1m-sgRNAs were heated at 75 °C for 5 min and cooled to room temperature for 15 min. S1m-sgRNA of 20 pmol was combined with streptavidin at 10:1, 1:1, and 1:10 molar ratios in a final volume of 5 μL and the mixture was allowed to complex for 10 min. The S1m-sgRNA-streptavidin complexes were run on a 1% agarose gel. Tertiary complexes were assembled by first mixing 15 pmol each of S1m-sgRNA and streptavidin. To this mixture, 6, 15, or 30 pmol of ssODN was added prior to running the complexes through a 1% agarose gel. All gels were run using Kb + ladder (Invitrogen) as a molecular weight marker to allow for inter-gel size comparisons even when running RNA samples. Uncropped gels can be found in Supplementary Information.

**Biotin competition assay.** S1m-sgRNA was heated to 75 °C for 5 min and cooled to room temperature. 20 pmol of each S1m-sgRNA and streptavidin were complexed for 10 min. Biotin of 80 pmol was added at 30, 20, 10, 5, and 0 min intervals prior to running the complexes through a 1% agarose gel.

**RNP delivery.** HEK transfections were performed using TransIT-X2 delivery system (Mirus Bio, Madison, WI) according to manufacturer's protocol. $2.5 \times 10^5$ cells per cm$^2$ were seeded in a 24-well plate 24 h prior to transfection. RNP complexes were formed in 25 μL of Opti-MEM (Life Technologies). 5 pmol of Cas9 protein, 6 pmol sgRNA, 5 pmol streptavidin, and 6 pmol ssODN were used. In a separate tube, 25 μL of Opti-MEM was combined with 0.75 μL of TransIT-X2 reagent and allowed to combine for 5 min. TransIT-X2 and RNP solutions were then mixed by gentle pipetting and placed aside for 15 min. After this incubation, 50 μL of solution were added dropwise into the well. Media was changed 24 h post transfection.

For HEK transfections involving quantum dots, Lipofectamine 2000 (Life Technologies) was used for delivery. Qdot-RNP complexes were formed using the following amounts of reagents for each well of a 24 well plate: 5 pmol of Cas9 protein, 6 pmol sgRNA, 5 pmol streptavidin, 3.125 pmol of quantum dots, and 3 μL lipofectamine per well. a quarter of these amounts were used when transfecting 5000 cells in 96 well plates.

All hPSC transfections were performed using the 4D-Nucleofector System (Lonza) in P3 solution using protocol CB-150. Cells were pretreated with Rho-kinase (ROCK) inhibitor (Y-27632 Selleck Chemicals) 24 h prior to transfection. 50 pmol of Cas9, 60 pmol sgRNA, 50 pmol streptavidin, and 60 pmol ssODN were used to form particles for ssODN-S1mplex as described above. Cells were then harvested using TrypLE (Life Technologies) and counted. A total of $2 \times 10^5$ cells per transfection were then centrifuged at 100g for 3 min. Excess media was aspirated and cells were resuspended using 20 μL of RNP solution per condition. After nucleofection, samples were incubated in nucleocuvettes at room temperature for 15 min prior to plating into one well of a 6-well plate containing E8 media + 10 μM ROCK inhibitor. Media was changed 24 h post transfection and replaced with E8 medium.

**Dynamic light scattering.** DLS was performed using a DynaPro NanoStar (Wyatt Technology) using small volume (4 μL) disposable cuvettes. 10 μg of each component was added into the cuvette and diluted as necessary with dH$_2$O to reach 4 μL of solution volume. Excess sgRNA was unable to be detected by DLS and was included to ensure all important species were able to complex together. In mixed component conditions, components were allowed to mix for 5 min while taking readings. Acquisitions were performed for 20 s with a minimum of four acquisitions per measurement. Five measurements were performed per sample and were conducted at room temperature. Data was graphed as a function of percent intensity.

**Immunocytochemistry.** To measure colocalization of S1mplex components, hPSCs were transfected with Cas9 protein and streptavidin-AF-647. Twenty-four

hours post transfection, cells were fixed using 4% PFA and incubated at room temperature for 10 min. Cells were then permeabilized using 0.05% Triton X-100 and incubated for 10 min. Following two washes with 5% goat serum, Cas9 antibody (Clontech #632607, 1:150 dilution) was added to cells and incubated overnight at 4 °C. The next day, cells were rinsed twice with 5% goat serum and then incubated with a goat anti-rabbit secondary antibody (Santa Cruz Biotech #sc-362262, 1:500 dilution) for 1 h at room temperature. Cells were then washed twice with PBS and mounted for imaging.

To visualize S1mplexes in the nucleus, HEK cells (HEK293T) were plated at 16 000 cells per well in an eight-well chamber slide at day 0. On day 1, 20 μL of transfection media was added to cells in 200 μL of maintenance media. Transfection media contained 20 μL Opti-MEM (Life Technologies), 10 pmol streptavidin Alexa Fluor 488 conjugate (Thermo Fisher), and 0.6 μL TransIT transfection reagent (Mirus). On day 3, cells were incubated with 1× CellMask plasma membrane stain (Thermo Fisher) and 1× Hoechst 33342 for 10 min. Following incubation at 37 °C, cells were immediately washed with PBS, and fixed in 4% paraformaldehyde (IBI Scientific) at room temperature for 15 min. Cells were analyzed using a Nikon Eclipse TI epifluorescent microscope and a Nikon AR1 confocal microscope.

**Multispectral imaging flow cytometry.** hPSCs were transfected and stained as described above. After staining, cells were centrifuged and resuspended in 50 μL PBS. Fluorescence was detected on ImageStream X Mark II (EMD Millipore) according to manufacture's instructions. Cellular co-localization was measured by IDEAS software package (Amnis) using the co-localization wizard.

**Quantum dot biotin conjugation.** To make Qdot-SS-S1mplexes, amine-PEG green fluorescent quantum dots (Qdot ITK 525,Thermo Fisher) were reacted with a degradable dithiol biotin linker (EZ-Link sulfo-NHS-biotin, Thermo Fisher) as follows: First, 25 μL of an 8 μM Quantum dot solution in 50 mM Borate buffer was desalted into PBS using Zeba desalting columns (40 K molecular weight cut off, Thermo Fisher) and then reacted with excess sulfoNHS-dithiol-biotin linker for 2 h at 4 °C with shaking. The conjugate was purified from excess linker through buffer exchange in the desalting columns. Quantum dots retained their fluorescence and were stored at 4 °C until use.

**Flow cytometry.** Flow cytometry of BFP expression and conversion to GFP was measured using a BD FACS Aria using the DAPI and FITC filters and analyzed using FlowJo. Voltages were established by running wild-type WA09 hPSCs as well as WA09-BFP hPSCs. Sorting was performed on a BD FACS Aria II with a nozzle size of 100 μm at room temperature. Cells were sorted into culture media.

**Genomic analysis.** DNA was isolated from cells using DNA QuickExtract (Epicentre, Madison, WI) following treatment by 0.05% trypsin-EDTA and centrifugation. The DNA extract solution was incubated at 65 °C for 15 min, 68 °C for 15 min, and finally 98 °C for 10 min. Genomic PCR was performed following manufacturer's instructions using AccuPrime HiFi Taq (Life Technologies) and 500 ng of genomic DNA. Products were then purified using AMPure XP magnetic bead purification kit (Beckman Coulter) and quantified using a Nanodrop2000. Primer sequences are listed in Supplementary Tables 3, 6, and 7. Genomic target sequences of predicted off-target sites are listed in Supplementary Table 5. For deep sequencing, samples were pooled and run on an Illumina HiSeq2500 high throughput at a run length of 2 × 125 bp or an Illumina Miseq 2 × 150 bp.

**Deep-sequencing data analysis.** A custom python script was developed to perform sequence analysis. The pipeline starts with preprocessing, which consists of filtering out low-quality sequences and finding the defined ends of the reads. For each sample, sequences with frequency of <10 were filtered from the data. Sequences in which the reads matched with primer and reverse complement subsequences classified as "target sequences." Target sequences were aligned with corresponding wild-type sequence using global pairwise sequence alignment. Sequences that were misaligned around the expected cut site were classified as NHEJ events while sequences that matched the ssODN template were classified as HDR events. The frequency, length, and position of matches, insertions, deletions, and mismatches were all tracked in the resulting aligned sequences. The custom script is available upon request.

**Statistics.** All error bars are shown as ±1 s.d. $p$ values were computed using a Student's two-tailed $t$ test and deemed significant at $\alpha < 0.05$.

**Data availability.** Raw reads and traces from sequencing are available at NCBI Bioproject PRJNA381066. Analysis script can be obtained at https://github.com/jaredc-s/BarcodingNGSSequencing

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

## Acknowledgements

We thank members of the Saha lab for helpful discussion and comments on the manuscript. We also thank plasmid depositors to Addgene, ArtforScience, James Thomson lab for use of their BD FACS Aria, the University of Wisconsin Biotechnology Center, and the University of Wisconsin Carbone Cancer Center Flow Cytometry Laboratory for technical support. We thank Aldevron and Mirus Bio for supplying reagents and technical support. We further thank M. Suzuki and T. Kamp for sharing Pompe hiPSCs. We acknowledge generous financial support from the National Science Foundation (CBET-1350178 and CBET-1645123), National Institute for Health (1R35GM119644-01), Environmental Protection Agency (EPA-G2013 –STAR-L1), Burroughs Wellcome Fund (Innovation in Regulatory Science Award), Wisconsin Alumni Research Foundation, and the Wisconsin Institute for Discovery.

## Author contributions

J.C.-S. and A.A.A. planned the research and analyzed the data. J.C.-S., A.A.A., and K.S. designed the experiments. J.C.-S., A.A.A., L.K, M.G., K.M., and M.L. performed the experiments. J.C.-S. and A.A.A wrote the manuscript with input from all the authors. K. S. supervised the research.

## Additional information

**Competing interests:** J.C.-S., A.A.A., L.K., and K.S. have filed a patent application on this work. The remaining authors declare no competing finanical interests.

