## [Peer Review File · Nature Communications]

Reviewers' comments:

Reviewer #1 (Remarks to the Author):

Carlson-Stevermer et al. report results of a novel approach to enhancing homology-directed repair (HDR) following CRISPR-Cas9 cleavage. They add to the sgRNA an RNA aptamer that binds streptavidin, then provide a biotin-labeled donor oligonucleotide. As streptavidin is expected to form a tetramer, this could bring up to 4 donor molecules to the specific target site. The authors report impressive increases in the ratio of HDR products to indels at two different targets in human cells: a synthetic BFP gene and a site in the endogenous EMX1 locus.

The chief problem with this manuscript is that the absolute levels of genome editing – and particularly of HDR – are very low compared to experience in other labs. In the best cases, these levels are in the range of 1% of all targets, and often they are well below 0.1% (which is 1000 reads per million; Supp. Figs. 4, 6). These frequencies would be unusable for most applications, especially when biallelic edits are required. Many other groups have achieved much higher HDR frequencies with standard reagents, including untethered ssODNs. For example, Richardson et al. (cited as ref. 25) were working in the range of 10-30% HDR, with even higher levels obtained with optimized donor designs. That group utilized HEK293T cells, as does the current study. To reiterate, absolute frequencies of editing are much more important than HDR/indel ratios for most applications.

Another issue is that no optimization, or even testing, of important variables is presented. Because of constraints imposed on the ssODN by physical tethering to the RNP complex, one would expect some exploration of the length, polarity, and mutation location. The gel shift data do not include reports of stoichiometries. For example, in Supp. Fig. 1c, what are the ratios of streptavidin to S1m-sgRNA? And in Figs. 1c and Supp. 2, what are the ratios of biotin-ssODN to the other components? What is the direct evidence that the shifted bands contain the components suggested? The degree that S1m-sgRNA is shifted by streptavidin seems rather too little (although I don't know exactly what to expect), and the attribution of the multiple shifted bands in Supp. Fig. 1c should be validated.

There are multiple other issues that I will simply list:

1. In several cases, there appears to be "editing" in untreated cells. This includes BFP loss in Supp. Fig. 4e and f; and BFP HDR (and NHEJ) in double-negative cells in Supp. Fig. 6.
2. In Supp. Fig. 6b, there seems to be an anomalously low yield of NHEJ products in double-negative cells. Are some of these anomalies due to the very low number of sequences in each class or to misattribution?
3. Molar amounts of all components should be given in the Methods, when describing how RNPs were assembled and mixed with ssODNs. It looks to me like smaller amounts of RNP were used than in some previous studies, and this could contribute to the very low editing frequencies.
4. No methods are provided for the gel shift studies.
5. It was sometimes unclear which cells were being used. On p. 4 in the top paragraph, there seems to be an abrupt switch from HEK to hPSCs. Only hESCs are described in the Methods. Are these the same as hPSCs?
6. In the experiments reported in Fig. 2, what proportion of cells took up quantum dots?
7. In the legend to Fig. 1c, what are "elongated bands"?
8. In Supp. Fig. 1a, the RNA should be shown with U, not T, and the source of the framework without the extra nucleotides in green should be given. If this paper is ultimately published, I think this diagram, or something like it, should appear with the main text.

Reviewer #2 (Remarks to the Author):

Increasing the frequency of homology-directed repair (HDR) has been of great interest since the inception of gene editing and especially so since the introduction of CRISPR. In this submission the authors use gRNAs containing streptavidin aptamers to build CRISPR complexes containing oligo templates for HDR and quantum dots for FACS. Using these complexes (dubbed "S1plexes" by the authors) enabled significant increases in HDR at the expense of error-prone NHEJ repair. This technology should be of interest to the expanding CRISPR community and the manuscript is solid.

I have a number of suggestions to improve the manuscript, in order of importance:

1. How the tethering of ssDNA oligos to the CRISPR RNPs promotes more efficient HDR is not clear. To start to address this question some variations on the basic S1plex format could quickly be tested. These could include using engineered monomeric streptavidin (see papers from the Sheldon Park lab) instead of the natural streptavidin tetramer, using a donor oligo targeting the complementary strand, and biotinylating the 3' end of the donor oligo, for example. This would provide important information on stoichiometry and orientation effects.

2. The authors found that inserting the streptavidin aptamers reduced the efficiency of cleavage using these gRNAs. Additional RNA sequences have been found to be tolerated at many different insertion points in gRNAs (e.g. see Shechner et al. 2015 Nature Methods). Have the authors tried any additional modes of insertion for the S1m stem loop? This could improve gRNA functionality. Also have the authors considered that the impaired gRNA activity could assist HDR? Slower Cas9 cutting might provide more opportunity for donor-dependent HDR compared to NHEJ. Testing gRNAs with different efficiencies/different S1m insertion points could help understand this and assist applications of this method.

3. Page 4, the HDR of 3 bp in the BFP gene was improved 18.4-fold using S1plexes but the EMX 12 bp HDR was only improved by 2.7-fold. Why wouldn't this technology improve all HDRs to a similar extent? After all, the ratio takes into account the lower efficiency of longer mutations repaired by the untethered oligo donor. Part of the problem here is that the examination of efficiencies of HDRs of different sequence lengths using the same target gene (BFP or EMX) and gRNA but differing only in the number of bp mutated was performed in either HEK (Figure 1d,e) or hPSCs (Figure 1 f,g) but not both, so they can't be directly compared. This information would be very valuable to help understand the design considerations for those choosing to use this approach.

4. In the DLS experiment shown in Figure 1b, what ratios were used for each component in the mixes? Does the almost complete binding of streptavidin within the S1plex occur at a 1:1 ratio? In the gel shown in Supplementary Figure 1c only 40% of the gRNA appeared to bind streptavidin. The authors should clarify this point.

5. The manuscript title is very vague and doesn't capture the key advance of the paper well. The authors' line on page 2 comes closer: "preassembling RNPs with donor template or other moieties that enable the isolation of precisely-edited cells". Mentioning some or all of "aptamers", "HDR" and "quantum dots" might be appropriate.

6. The manuscript is formatted in a brief communication style. The authors should add section titles and move the Supplementary Text and Supplementary References into the main manuscript, along with some of the Supplementary Figures (e.g. Supplementary Figure 1a, the full gel in Supplementary Figure 2 in place of Figure 1c, Supplementary Figures 4 and 5, Supplementary Figure 6a-c).

7. A recent paper from Ma et al. in Cell Reports is mentioned in passing at the end of page 6. It would be good to compare and contrast this approach, using a Cas9-avidin fusion, with that of the authors.

8. Pages 6 and 8, "complexation" is not a word.
9. The abbreviation "Sp.Cas9" is more frequently used as "SpCas9" in the recent literature.
10. "5' end", a prime symbol should be used instead of an apostrophe.
11. Page 3, "create a loss of mCherry fluorescence" should be "result in a loss of mCherry fluorescence" or similar.

Responses to reviewer's comments are highlighted in red.

Reviewer #1 (Remarks to the Author):

Carlson-Stevermer et al. report results of a novel approach to enhancing homology-directed repair (HDR) following CRISPR-Cas9 cleavage. They add to the sgRNA an RNA aptamer that binds streptavidin, then provide a biotin-labeled donor oligonucleotide. As streptavidin is expected to form a tetramer, this could bring up to 4 donor molecules to the specific target site. The authors report impressive increases in the ratio of HDR products to indels at two different targets in human cells: a synthetic BFP gene and a site in the endogenous EMX1 locus.

1) The chief problem with this manuscript is that the absolute levels of genome editing – and particularly of HDR – are very low compared to experience in other labs. In the best cases, these levels are in the range of 1% of all targets, and often they are well below 0.1% (which is 1000 reads per million; Supp. Figs. 4, 6). These frequencies would be unusable for most applications, especially when biallelic edits are required. Many other groups have achieved much higher HDR frequencies with standard reagents, including untethered ssODNs. For example, Richardson et al. (cited as ref. 25) were working in the range of 10-30% HDR, with even higher levels obtained with optimized donor designs. That group utilized HEK293T cells, as does the current study. To reiterate, absolute frequencies of editing are much more important than HDR/indel ratios for most applications.

We thank the reviewer for raising this important point. The absolute levels of editing in our study are consistent with recent reports that characterize efficiency within the entire cell population after Cas9 RNP delivery: two recent reports with human iPSCs reported ~1-10% of alleles being edited (A. M. Tidball et. al. *Stem Cell Reports*, 2017 and H. Ma et. al. *Nature*, 548, 413–419, 2017). New deep sequencing experiments with our S1plexes at the *BFP* locus and a third locus, *GAA*, with <5 bp changes confirm our consistent 1-10% levels of absolute editing (please see **Figure 4**). Also, we now explicitly state the absolute levels of editing on page 6 in the main text (noted in red). Many studies to date perform selection (subcloning, drug selection or flow cytometry) which is typically not accounted for in the efficiency calculations reported in literature. Our S1plex strategy should be complementary to those enrichment and selection approaches.

2) Another issue is that no optimization, or even testing, of important variables is presented. Because of constraints imposed on the ssODN by physical tethering to the RNP complex, one would expect some exploration of the length, polarity, and mutation location.

We have now explored these issues over the last three months with significant new experimentation. New data in **Figures 3a** and **4** explore the effects of various ssODN lengths, ssODN polarities, ssODN tethering locations, and S1m aptamer locations. A detailed discussion of these results is on pages 9-10 of the main text (additions are noted in red).

3) The gel shift data do not include reports of stoichiometries. For example, in Supp. Fig. 1c, what are the ratios of streptavidin to S1m-sgRNA? And in Figs. 1c and Supp. 2, what are the ratios of biotin-ssODN to the other components? What is the direct evidence that the shifted bands contain the components suggested?

We have now updated the Methods on page 12 as well as the main text (noted in red) to include the relative stoichiometry of components in the electrophoretic mobility shift assays. Further, we have now added data on the localization of components within human PSCs in Figure 2b-e. Through the use of a

stringent control where only the S1m aptamer is absent, these results - together with the dynamic light scattering results in **Figure 2a** - demonstrate that the components assemble as stated.

4) The degree that S1m-sgRNA is shifted by streptavidin seems rather too little (although I don't know exactly what to expect), and the attribution of the multiple shifted bands in Supp. Fig. 1c should be validated.

We agree with the reviewer that that the shift appears small, but the magnitude of the shift is hard to predict. Electrophoretic mobility shifts of proteins are dependent the charge and pore size of the agarose gel, which makes it is difficult to predict exactly the shift of proteins with complex native structures. Given the new data in **Figure 2** noted above in response to point 3), the EMSA shift from streptavidin relative to the appropriate controls indicates robust assembly of the S1mplex.

5) There are multiple other issues that I will simply list:

1. In several cases, there appears to be "editing" in untreated cells. This includes BFP loss in Supp. Fig. 4e and f; and BFP HDR (and NHEJ) in double-negative cells in Supp. Fig. 6.

We thank the reviewer for highlighting this point. The BFP transgene reporter system is an imprecise measure of gene editing, and therefore careful measures have been taken to exclude cells from analysis that have lost fluorescence through standard culturing. As now noted in the Methods on page 11 of the main text (noted in red), we performed routine sorting of the cell line to maintain a pure population. For the BFP experiments, comparisons between flow cytometry and deep sequencing reported **Supplemental Figure 4** and discussed in the Supplementary Text.

6) In Supp. Fig. 6b, there seems to be an anomalously low yield of NHEJ products in double-negative cells. Are some of these anomalies due to the very low number of sequences in each class or to misattribution?

We thank the reviewer for pointing out this possibility. We reanalyzed the deep sequencing data with different criteria for errors in sequencing. Through this analysis, we identified additional sequence reads that were not attributed to NHEJ products. We have now updated **Figure 6** and **Supplementary Figure 6** with our new analysis.

7) Molar amounts of all components should be given in the Methods, when describing how RNPs were assembled and mixed with ssODNs. It looks to me like smaller amounts of RNP were used than in some previous studies, and this could contribute to the very low editing frequencies.

We updated the Methods on pages 11-12 of the main text (noted in red) with molar quantities. Lower concentrations are used in single S1mplex experiments than some previous studies, so that multiplexed S1mplex experiments in **Figure 6** could be compared to the experiments in published Richardson et. al. *Nature Biotechnology*, 2016 and Liang et. al. *Journal of Biotechnology*, 2017.

8) No methods are provided for the gel shift studies.

We apologize for this oversight and have now provided Methods on page 12 for the gel shift assays.

9) It was sometimes unclear which cells were being used. On p. 4 in the top paragraph, there seems to be an abrupt switch from HEK to hPSCs. Only hESCs are described in the Methods. Are these the same as hPSCs?

Clarifications as to what cell type was used for each experiment have been made more explicit both within the text and figures. Methods have been updated to reflect these changes as well.

10) In the experiments reported in Fig. 2, what proportion of cells took up quantum dots?

We now include the flow cytometry data in **Figure 5d** after Qdot-S1mplex delivery. Nearly all cells take up quantum dots but to varying degrees. For gating 'positive' populations, we set up gates that try to exclude as many of untransfected population as possible while getting 8-12% of the cells for further downstream analysis. In the representative flow histogram in Figure 5, the sorting gate is indicated.

11) In the legend to Fig. 1c, what are “elongated bands”?

In our first submission, elongated bands referred to bands that consisted of S1m and streptavidin. We have removed this language in the current text and have more explicitly described the various species in our EMSA.

12) In Supp. Fig. 1a, the RNA should be shown with U, not T, and the source of the framework without the extra nucleotides in green should be given. If this paper is ultimately published, I think this diagram, or something like it, should appear with the main text.

We thank the reviewer for their detailed comments on our recent work and have updated the diagram in **Figure 1b** to reflect this important distinction.

Reviewer #2 (Remarks to the Author):

Increasing the frequency of homology-directed repair (HDR) has been of great interest since the inception of gene editing and especial so since the introduction of CRISPR. In this submission the authors use gRNAs containing streptavidin aptamers to build CRISPR complexes containing oligo templates for HDR and quantum dots for FACS. Using these complexes (dubbed “S1mplexes” by the authors) enabled significant increases in HDR at the expense of error-prone NHEJ repair. This technology should be of interest to the expanding CRISPR community and the manuscript is solid.

I have a number of suggestions to improve the manuscript, in order of importance:

13) 1. How the tethering of ssDNA oligos to the CRISPR RNPs promotes more efficient HDR is not clear. To start to address this question some variations on the basic S1mplex format could quickly be tested. These could include using engineered monomeric streptavidin (see papers from the Sheldon Park lab) instead of the natural streptavidin tetramer, using a donor oligo targeting the complementary strand, and biotinylating the 3' end of the donor oligo, for example. This would provide important information on stoichiometry and orientation effects.

We thank the reviewer for these important considerations and have now performed additional experiments to explore the effects of various ssODNs. New deep sequencing data in Figure 4 now shows changes in the HDR:indel ratio at the *BFP* locus when various ssODNs are used. In addition, we tested various ssODNs in new experiments by targeting a third locus, *GAA* (**Figure 4**). Results from these experiments are discussed in the main text on pages 9-10 (noted in red).

14) The authors found that inserting the streptavidin aptamers reduced the efficiency of cleavage using these gRNAs. Additional RNA sequences have been found to be tolerated at many different insertion points in gRNAs (e.g. see Shechner et al. 2015 Nature Methods). Have the authors tried any additional modes of insertion for the S1m stem loop? This could improve gRNA functionality. Also have the authors considered that the impaired gRNA activity could assist HDR? Slower Cas9 cutting might provide more opportunity for donor-dependent HDR compared to NHEJ. Testing gRNAs with different efficiencies/different S1m insertion points could help understand this and assist applications of this method.

The reviewer brings up an important design consideration for the S1mplex strategy. We created two additional S1m-sgRNAs that reflect different regions to place the aptamer. Gene editing with the three S1m-sgRNA variants are now presented in **Figures 1 and 2**. Results from these experiments are discussed in the main text on page 9 (noted in red). While we agree that slower cutting from S1m-sgRNA

RNPs could make a difference, the difference in **Figure 3e** for instance when streptavidin is added indicates that assembly of the S1mplex is critical to increasing HDR:indel ratios.

15) Page 4, the HDR of 3 bp in the BFP gene was improved 18.4-fold using S1mplexes but the EMX 12 bp HDR was only improved by 2.7-fold. Why wouldn't this technology improve all HDRs to a similar extent? After all, the ratio takes into account the lower efficiency of longer mutations repaired by the untethered oligo donor. Part of the problem here is that the examination of efficiencies of HDRs of different sequence lengths using the same target gene (BFP or EMX) and gRNA but differing only in the number of bp mutated was performed in either HEK (Figure 1d,e) or hPSCs (Figure 1 f,g) but not both, so they can't be directly compared. This information would be very valuable to help understand the design considerations for those choosing to use this approach.

We thank the reviewer for this important point. The HDR:Indel ratio increased to a lesser extent at some alleles may be due to a couple of factors. First, the length of the homology arms decreased with longer insertions. Second, when editing transgene or mutant alleles, there was no second exact allele that could be used for HDR of the DSB. Finally, chromatin dynamics may play a role in the repair at these alleles (actively transcribed vs. silenced). Due to these considerations, we have always used a control transfection where S1m is absent, so that conclusions on the gene editing activity of the S1mplex could be made without necessarily comparing across loci or across different types of insertion. We have noted these points in the discussion on pages 9-10 of the main text (noted in red).

16) In the DLS experiment shown in Figure 1b, what ratios were used for each component in the mixes? Does the almost complete binding of streptavidin within the S1mplex occur at a 1:1 ratio? In the gel shown in Supplementary Figure 1c only 40% of the gRNA appeared to bind streptavidin. The authors should clarify this point.

Additional detail has been added regarding dynamic light scattering traces of the S1mplex. 10 μ g of each component was mixed in order to provide a signal that could be detected by the DLS instrument. While not a molar equivalence, free sgRNAs were added in excess to promote binding. These free sgRNAs are not detected by the DLS instrument. Molar excess of sgRNA was not used in the gel assay, so lower binding is expected in that assay. We have noted these points in the Methods on page 13.

17) The manuscript title is very vague and doesn't capture the key advance of the paper well. The authors' line on page 2 comes closer: "preassembling RNPs with donor template or other moieties that enable the isolation of precisely-edited cells". Mentioning some or all of "aptamers", "HDR" and "quantum dots" might be appropriate.

We thank the reviewer for the suggestions. We have updated the title of the manuscript accordingly.

18) The manuscript is formatted in a brief communication style. The authors should add section titles and move the Supplementary Text and Supplementary References into the main manuscript, along with some of the Supplementary Figures (e.g. Supplementary Figure 1a, the full gel in Supplementary Figure 2 in place of Figure 1c, Supplementary Figures 4 and 5, Supplementary Figure 6a-c).

We have rewritten the manuscript to conform to these style guidelines.

19) A recent paper from Ma et al. in Cell Reports is mentioned in passing at the end of page 6. It would be good to compare and contrast this approach, using a Cas9-avidin fusion, with that of the authors.

We have expanded our discussion of this recent paper on page 10 of the main text (noted in red).

20) Pages 6 and 8, "complexation" is not a word.

We have removed the word complexation from the manuscript.

21) The abbreviation “Sp.Cas9” is more frequently used as “SpCas9” in the recent literature.

We thank the reviewer for this suggestion. This change has been made.

10. “5’ end”, a prime symbol should be used instead of an apostrophe.

We have made this substitution.

11. Page 3, “create a loss of mCherry fluorescence” should be “result in a loss of mCherry fluorescence” or similar.

We have corrected this language.

Reviewers' comments:

Reviewer #1 (Remarks to the Author):

My principal complaint about this manuscript still stands. Despite the claim to the contrary in the response from the authors, absolute values of HDR and NHEJ outcomes are provided essentially only in Supplementary Figure 4, where they are reported to be in the range of 0.1-1.0% of all cells. It is also clear in that figure that the absolute value of HDR decreases with the modified sgRNA compared to standard sgRNA. The reported frequencies are too low to be of general use, so the improved HDR:NHEJ ratios that dominate other figures do not have much impact.

There are a number of apparent inconsistencies and peculiarities in the manuscript, of which I note only a few:

- 1) The diagrams in Figure 1b do not depict sgRNA structure accurately. The natural stem loops should be shown explicitly, including the cases in which they are interrupted. Both the SL2 and SL3 RNAs have long stems that come neither from standard sgRNA nor the S1m aptamer. This should be acknowledged. Finally, the insertion in the SL3 construct is not in stem-loop 3, but follows that feature and is really an addition to the 3' end.
- 2) Figure 2a indicates that addition of streptavidin shifts all of the Cas9-S1m-sgRNA complex, while multiple other experiments show that the binding of streptavidin to this RNA is quite weak (Supp. Figs. 1b, 3b, 3c).
- 3) Several controls are missing from Supp. Fig. 3b that are necessary to identify individual bands: S1m-sgRNA alone; S1m-sgRNA + streptavidin, but without Biotin-ssODN; Biotin-ssODN + streptavidin (without sgRNA).
- 4) I have been reliably informed that the Amnis ImageStreamX Mark II Imaging Flow Cytometer does not produce phase contrast images, so it is puzzling where the pictures in Figure 2b come from.

Reviewer #2 (Remarks to the Author):

In this revised submission the authors have addressed the key concerns raised in the first round of review and I believe this paper will be of general interest to the gene editing community. Therefore I recommend publication.

Responses to reviewer's comments are highlighted in red.

Reviewers' comments:

Reviewer #1 (Remarks to the Author):

My principal complaint about this manuscript still stands. Despite the claim to the contrary in the response from the authors, absolute values of HDR and NHEJ outcomes are provided essentially only in Supplementary Figure 4, where they are reported to be in the range of 0.1-1.0% of all cells.

We appreciate the concern over absolute levels of editing and apologize for any confusion regarding these levels in our manuscript. We now present the absolute levels of HDR and NHEJ in Figure 4 of the Main Text, as shown below. Absolute HDR levels are routinely above 1% and can reach up to 6.6% using the S1mplex strategy within hPSCs. These graphs now provide the reader added detail on the absolute levels of editing in these key experiments. Further, we have provided more detail with the standard error across replicates regarding absolute editing within the Main Text (modifications are highlighted in red).

In tabular form, the absolute editing levels from all experiments are listed in Supplementary Table 9 (next page) and the raw sequences will be accessible on Bioproject PRJNA381066 upon publication.

Supplementary Table 9: Absolute levels of editing for each experimental replicate. Analysis of HDR and NHEJ rates following deep sequencing in each experimental condition and replicate presented in this work.

Experiment	Condition	Cell Line	Locus	Number of base pair changes	HDR1	HDR2	HDR3	NHEJ1	NHEJ2	NHEJ3
Figure 3, also Supplemen- tary Figure 4	ssODN-S1m-sgRNA-1 S1mplex	hPSC	BFP	3	0.49%			0.43%		
	ssODN-S1m- sgRNA-2 S1mplex	hPSC	BFP	3	0.76%			0.68%		
	ssODN-S1m- sgRNA-3 S1mplex	hPSC	BFP	3	0.30%			0.35%		
	Standard sgRNA ssODN	hPSC	BFP	3	1.3%			3.9%		
	Standard sgRNA ssODN	HEK	BFP	3	0.52%			2.2%		
	ssODN-S1mplex Standard sgRNA	HEK	BFP	3	0.99%			0.23%		
	ssODN ssODN-S1mplex	HEK	EMX1	12	1.0%			3.9%		
	ssODN-S1mplex Standard sgRNA	HEK	EMX1	12	0.07%			0.09%		
	ssODN	hPSC	BFP	18	0.045%			11%		
	sgRNA Bio-ssODN	hPSC	BFP	18	0.002%			9.8%		
	S1m Bio-ssODN (-SA)	hPSC	BFP	18	0.009%			1.3%		
	ssODN-S1mplex	hPSC	BFP	18	0.079%			2.0%		
	S1m Bio-ssODN (-SA)	hPSC	EMX1	18	0.001%			1.8%		
	ssODN-S1mplex	hPSC	EMX1	18	0.052%			6.0%		
	Figure 4	5' -67 NonPAM	hPSC	BFP	3	1.0%	1.1%	3.6%	0.51%	0.22%
3' -67 NonPAM		hPSC	BFP	3	1.0%	0.97%	5.9%	0.18%	0.15%	1.7%
5' -30 NonPAM		hPSC	BFP	3	1.1%	1.2%	3.1%	0.11%	0.30%	1.4%
3' -30 NonPAM		hPSC	BFP	3	1.0%	1.3%	3.4%	0.31%	0.44%	1.5%
5' -67 PAM		hPSC	BFP	3	1.3%	1.1%	2.8%	0.37%	0.13%	1.4%
3' -67 PAM		hPSC	BFP	3	1.0%	1.2%	5.0%	0.34%	0.34%	2.0%
5' -30 PAM		hPSC	BFP	3	1.1%	1.1%	6.6%	0.27%	0.25%	1.4%
3' -30 PAM		hPSC	BFP	3	1.0%	1.0%	3.4%	0.20%	0.24%	1.4%
RNP Control		hPSC	BFP	3	3.1%	1.1%	1.4%	2.9%	2.4%	4.6%
RNP Control		hPSC	GAA	2	1.7%	2.7%		2.2%	3.7%	
3' -34 PAM		hPSC	GAA	2	1.2%	3.8%		0.18%	0.46%	
5' -34 PAM		hPSC	GAA	2	1.4%	2.6%		0.40%	0.37%	
3' -34 NonPAM		hPSC	GAA	2	1.1%	1.0%		0.17%	0.33%	
5' -34 NonPAM		hPSC	GAA	2	1.1%	1.1%		0.18%	0.27%	
Figure 6d, also Supplemen- tary Figure 6b Multiplexed editing for both EMX and BFP		Double Negative	HEK	EMX1	12	0.02%			0.04%	
	Green+	HEK	EMX1	12	0.15%			0.36%		
	Red+	HEK	EMX1	12	0.02%			0.04%		
	Double Positive	HEK	EMX1	12	0.10%			0.13%		
	Double Negative	HEK	BFP	3	0.29%			0.05%		
	Green+	HEK	BFP	3	0.11%			0.02%		
	Red+	HEK	BFP	3	0.52%			0.01%		
Double Positive	HEK	BFP	3	0.57%			0.04%			

Further, we have now added Supplementary Table 8 that provides a summary of the absolute levels of editing above and those reported in literature.

Supplementary Table 8. Absolute levels of editing in this study and prior studies to date. Prior studies were selected based on a comparable number of base pair changes within the ssODN template and use of deep or Sanger sequencing of the edited locus. In our study, deep sequencing assays were performed shortly after RNP delivery without significant selection, while Sanger sequencing occurred after clonal isolation and expansion without drug selection.

Experiment	Cell type	Gene	# of base pair changes	% of HDR (absolute percent of deep sequencing reads)	% of NHEJ (absolute percent of deep sequencing reads)	Reference
Figure 3, averaged	HEK	BFP, EMX1	3, 12	0.53%	0.16 %	This study
Figure 3, averaged	hPSC	BFP, EMX1	18	0.34%	1.89%	This study
Figure 4, averaged	hPSC	BFP, GAA	2-3	2.0%	0.58%	This study
Figure 6, averaged	HEK	BFP and EMX1	3 and 12	0.34%	0.14%	This study
Averaged across experiments involving HDR as assayed by deep sequencing (Figs. 3, 4, 6)	HEK, hPSC	BFP, GAA, EMX1	2 to 18	1.6%	0.67%	This study
Small base pair changes	HEK, hPSC	BFP, GAA	<5	1.8%	0.75%	This study
HDR as assayed by Sanger sequencing of enriched clones (Fig. 6b)	HEK	BFP	3	24%	24%	This study
Figure 1	hPSC	SCN8A, SCN1B, CHD2, PCDH19, HPRT1, SMC1A	N/A	N/A	1-17.5%	A. M. Tidball et al. Stem Cell Reports, 2017
Extended data Figure 1f	hPSC	MYBPC3	4	1.5%	3.8%	H. Ma et al. Nature, 548, 413–419, 2017
Extended data Table 1	hPSC	APP, PSEN1	2	0.3-6.0%	Not reported	Paquet, D. et al. Nature 533, 125–129 (2016).
Figure 3a	hPSC	CCR5	2	0.2-1.6%	1-1.8%	Yang, L. et al. Nucl. Acids Res. (2013). doi:10.1093/nar/gkt555
Table S1	hPSC	AKT2	1	6.4%*	Not reported	Ding, Q. et al. Cell Stem Cell 12, 393–394 (2013).
Figure 2f	hPSC	SOD1	1	0.35-3.1%	27-32%	Yu, C. et al. Cell Stem Cell 16, 142–147 (2015).
Figure 5	HEK293T	RBM20, ATP7B	1	0.09-0.6%	3-9%	Miyaoaka, Y. et al. Scientific Reports 6, srep23549 (2016).
Figure 5	hPSC	RBM20	1	0.005-0.1%	1-10%	Miyaoaka, Y. et al. Scientific Reports 6, srep23549 (2016).
Figure 2c, Supplementary Table 12 before drug selection	HCT116	ERCC3	2	0.3-5.2%	7.1-12.6%	Smurnyy, Y. et al. Nat Chem Biol 10, 623–625 (2014).
Figure 2a, unexpanded cells	HSPC	HBB	1	6-11%	10-30%	DeWitt, M. A. et al. Science Translational Medicine 8, 360ra134- (2016).
Figure 4c	hPSC	EGFP	2	2.5-4.5%**	30-60%**	Howden, S. E. et al. Stem Cell Reports doi:10.1016/j.stemcr.2016.07.001
Supplemental Table S2	HSPC	CD45	2	24%	16%***	Gundry, M. C. et al. Cell Rep 17, 1453–1461, 2016
Figure 3c	HEK	BFP	3	9-63%	25%-90%	Richardson, C. D., Ray, G. J., DeWitt, M. A., Curie, G. L. & Corn, J. E. Nat Biotech 34,

						339–344, 2016
Figure S6, 7	HEK	EMX1	4	43-66%	2-31%****	Richardson, C. D., Ray, G. J., DeWitt, M. A., Curie, G. L. & Corn, J. E. Nat Biotech 34, 339–344, 2016
Figure 3	HEK	EMX1	12	0-6%*****	Not reported	Lin, S. et al. eLife 2014;3:e04766
Figure 4	hPSC	EMX1	12	0-1.6%	3-44%	Lin, S. et al. eLife 2014;3:e04766

* Enriched by flow cytometry for Cas9-GFP expressing cells; 10 of 94 clones with monoallelic mutation introduced and 1 of 94 clones with biallelic HDR editing, resulting in 12 out of 188 alleles = 6.4%

** With plasmid HDR donors; assayed by flow cytometry

***Only the most common indels are reported. Reported frequencies for all alleles add up to 85% in this table.

**** ssODN most similar to work involved in this paper induced 2-9% HDR and 54-55% NHEJ as measured by RFLP

***** ssODN similar to work in the paper was below limit of detection using RFLP assay.

All Cas9 studies involving ssODN-mediated HDR within hPSCs and the majority of studies in this field present HDR levels between ~0.1-10%. We have found only a few remarkable outliers: 10-60% within HEK cells (Richardson, C. D., Ray, G. J., DeWitt, M. A., Curie, G. L. & Corn, J. E. *Nat Biotech* 34, 339–344, 2016) and 24% within HSPCs (Gundry, M. C. *et al. Cell Rep* 17, 1453–1461, 2016). Nearly all other studies rely on selection to enrich for cells with HDR, and therefore efficiencies cannot be directly compared. Finally, absolute levels of HDR have been noted to vary across cell types and loci (Fellmann, C., Gowen, B. G., Lin, P.-C., Doudna, J. A. & Corn, J. E. Cornerstones of CRISPR-Cas in drug discovery and therapy. *Nat Rev Drug Discov* 16, 89–100 (2017). and Miyaoaka, Y. *et al.* Systematic quantification of HDR and NHEJ reveals effects of locus, nuclease, and cell type on genome-editing. *Scientific Reports* 6, srep23549 (2016). We have added a paragraph in the Discussion on these points.

2) It is also clear in that figure that the absolute value of HDR decreases with the modified sgRNA compared to standard sgRNA.

On average, the absolute value of HDR remains constant with the modified sgRNA within a ssODN-S1mplex (Figure 4a) compared to standard sgRNA conditions, while decreasing the levels of NHEJ (Figure 4b).

Therefore, the absolute value of HDR *does not* decrease with the modified sgRNA within a S1mplex, indicating that our strategy does not compromise the absolute levels of HDR in general.

In Supplementary Figure 4, the GFP gain in part d does show slight statistically insignificant decrease in HDR according to the flow cytometry assay (0.9 ± 0.5 vs. $0.3 \pm 0.5\%$ cells within FITC-A gates). These levels are highly dependent on the timing of the assay post-delivery, as well as the stringency of the gates and sensitivity of the flow cytometer. As noted in the Supplementary Text, deep sequencing is a more reliable and objective measure of HDR. For the deep sequencing results in Supplementary Figure 4, parts b, i and j clearly show higher absolute values of HDR when the modified sgRNA is used within a S1mplex. These absolute levels are shown now in Supplementary Table 9 (bolded is the ssODN-S1mplex, whereas the italicized sgRNA ssODN is the standard conditions):

Experiment	Condition	Cell Line	Locus	Number of base pair changes	HDR1	NHEJ1
Figure 3, also Supplementary Figure 4	sgRNA ssODN	HEK	BFP	3	0.50%	2.2%
	ssODN-S1mplex	HEK	BFP	3	0.84%	0.20%
	sgRNA ssODN	hPSC	BFP	18	0.045%	11%
	sgRNA Bio-ssODN	hPSC	BFP	18	0.002%	9.8%
	S1m Bio-ssODN (-SA)	hPSC	BFP	18	0.009%	1.3%
	ssODN-S1mplex	hPSC	BFP	18	0.079%	2.0%
	S1m Bio-ssODN (-SA)	hPSC	EMX1	18	0.001%	1.8%
	ssODN-S1mplex	hPSC	EMX1	18	0.052%	6.0%

There is a significant drop in HDR levels as we move from 3 to 18 base pair changes, which is to be expected due to shorter homology arms in the ssODN (Yang, L. et al. Optimization of scarless human stem cell genome editing. Nucl. Acids Res. (2013). doi:10.1093/nar/gkt555). However, absolute levels of HDR are at similar or higher levels than the standard sgRNA and ssODN controls (0.84% vs. 0.50% for 3 bp and 0.079% and 0.052% vs. 0.045% for the 18 bp changes). For the 18 bp change at the BFP locus, an additional control involving biotin-ssODN and a standard sgRNA (underlined) also indicates higher levels of HDR with the S1mplex (0.079% vs. 0.002%), so the modified sgRNA does not compromise HDR levels. For the 18 bp change at the *EMX1* locus, again a comparison using a similar control (underlined) indicates a high level of HDR (0.052% vs. 0.001%). Compared to the 18 nt change in BFP using standard sgRNA and ssODNs, similar HDR levels were observed (0.052% vs. 0.045%).

For a 12 bp change at *EMX1* within HEKs, we did see one significant drop in HDR with the S1mplex (0.07% vs 1%) for experiments in Figure 3c (with Supplementary Figure 4) and Figure 6 (with Supplementary Figure 6b):

Experiment	Condition	Cell Line	Locus	Number of base pair changes	HDR1	NHEJ1
Figure 3c and Supplementary Figure S4	sgRNA ssODN	HEK	EMX1	12	0.92%	3.1%
	ssODN-S1mplex	HEK	EMX1	12	0.07%	0.09%
Figure 6d, also Supplementary Figure 6b	Green+	HEK	EMX1	12	0.15%	0.36%
Multiplexed editing for both EMX and BFP	Double Positive	HEK	EMX1	12	0.10%	0.13%

While we do not have a mechanistic explanation for this drop at the *EMX1* locus with the 12 bp change, insertion at this locus has been reported to be more challenging than at *BFP* (Richardson *et al. Nature Biotechnology* 2016) and may be highly dependent on length of the homology arms surrounding the insertion (Lin *et al. eLife* 2014). We also note that a large 18 bp insertion at the BFP locus dropped significantly with the use of biotinylated ssODN and standard sgRNAs (0.002% as noted above), and that the drop at the *EMX1* locus with similar large base pair changes could be due to biotinylated ssODNs with large base pair changes, rather than the modified sgRNA alone. This exception is now noted in the Discussion on page XXX.

The key claim of the manuscript involving higher HDR:NHEJ ratios when using the full S1mplex is still consistent with editing at this locus at all of the tested conditions with 12 bp changes at the *EMX1* locus (unsorted, Green+ and Double Positive+). We could remove the experiments involving *EMX1* from the manuscript if deemed appropriate by the editorial team and Reviewer #1 due to the concerns over absolute HDR levels. However, we believe this level of transparency regarding variation from our experiments will be useful to the *Nature Communications* readership as they use the S1mplex tool.

3) The reported frequencies are too low to be of general use, so the improved HDR:NHEJ ratios that dominate other figures do not have much impact.

Absolute levels of 0.1-10% HDR is useful for a variety of studies involving mutation or single nucleotide polymorphisms (SNP) editing *in vitro* and *in vivo*. In our study, the lowest absolute levels were seen in attempting to make large 12-18 bp insertions with smaller homology arms within a constant length ssODN. The more common short (<5 bp) insertions had higher absolute

HDR levels (1.8%) and are likely to be used widely across the life sciences. These cases - as illustrated with 13 examples below - is where the S1mplex would likely have the most immediate impact.

For *in vitro* work, where more HDR would generate more precisely-edited cell lines, disease modeling and drug discovery would be impacted as noted other experts:

In neuroscience, a recent review describes such studies as an “*experimental approach that allows the study of the functional effects of disease-associated risk in complex disease by combining genome wide association studies (GWAS) and genome-scale epigenetic data to prioritize disease-associated risk variants with efficient gene editing technologies in human pluripotent stem cells (hPSCs).*” from Soldner, F. & Jaenisch, R. In Vitro Modeling of Complex Neurological Diseases. in *Genome Editing in Neurosciences* 1–19 (Springer, Cham, 2017). doi:10.1007/978-3-319-60192-2_1

An example of a high-impact study enabled by <7% HDR within hiPSCs is Soldner, F. et al. Parkinson-associated risk variant in distal enhancer of \$\alpha\$ -synuclein modulates target gene expression. *Nature* 533, 95–99 (2016).

In cancer, a recent review describes the potential impact of HDR within hiPSCs: “*Owing to the ease of single-cell subcloning, iPSCs are highly amenable to the introduction of precise genetic modifications by the CRISPR–Cas9 system or other genome-editing tools... [HDR can be used] to model hotspot point mutations in cancer-promoting genes. The CRISPR–Cas9 system thus enables modeling...of cancer mutations in their natural genomic context.*” from Papapetrou, E. P. Patient-derived induced pluripotent stem cells in cancer research and precision oncology. *Nat Med* 22, 1392–1401 (2016).

In drug discovery, Novartis used 0.3-5.2% HDR to knock-in alleles for drug target validation: “*Analysis of amplicon sequencing showed that before drug selection, the desired point mutations were present in 5.2% of alleles in the D54H locus and 0.3% of alleles in the F482 locus....CRISPR gene editing makes it easy to recapitulate desired mutations in a cell line of choice in both loss-of-function (6-TG) and change- of-function (triptolide) cases. Here we demonstrated the feasibility of this new target validation approach with a simple phenotypic selection by cellular toxicity. It is entirely conceivable that the same technology can be more broadly applied to nontoxic compounds...*” from Smurnyy, Y. et al. DNA sequencing and CRISPR-Cas9 gene editing for target validation in mammalian cells. *Nat Chem Biol* 10, 623–625 (2014).

For somatic editing work *in vivo*, 0.1-10% HDR could also have a high impact:

Less than 2% HDR *in vivo* (Figure 4h) generated tailored mouse models of cancer, and authors anticipate that, “*The ability to introduce covalent modifications in the genome of somatic cells enables the study of gene function for many areas of biology and provides significant time savings over conventional transgenic technologies.*” From Platt, R. J. et al. CRISPR-Cas9 Knockin Mice for Genome Editing and Cancer Modeling. *Cell* 159, 440–455 (2014).

A recent review of gene targeted therapeutic strategies for Duchenne muscular dystrophy (DMD) stated, “*Dystrophin levels as low as 3–15% of wild type are sufficient to ameliorate pathologic symptoms in the heart and skeletal muscle.*” from Tabebordbar, M., Cheng, J. & Wagers, A. J. Therapeutic Gene Editing in Muscles and Muscle Stem Cells. in *Genome*

Editing in Neurosciences 103–123 (Springer, Cham, 2017). doi:10.1007/978-3-319-60192-2_10

Phenotypic correction of haemophilia B can be achieved at 4-7% HDR as shown in Figure 5 of Li, H. et al. In vivo genome editing restores haemostasis in a mouse model of haemophilia. Nature 475, 217–221 (2011).

Even HDR within 0.1% of cells can rescue hereditary tyrosinemia type I as indicated by two papers:

Paulk, N. K. et al. Adeno-associated virus gene repair corrects a mouse model of hereditary tyrosinemia in vivo. Hepatology 51, 1200–1208 (2010). and Yin, H. et al. Genome editing with Cas9 in adult mice corrects a disease mutation and phenotype. Nat Biotech 32, 551–553 (2014).

Editas Medicine has active projects that involve ssODN-mediated HDR for an autologous hematopoietic stem/progenitor cells (HSPCs) editing strategy. They report *up to* 12% HDR (with a 69±8% NHEJ) in their studies [Heath, J. M. et al. Precise and Efficient Caspr/Cas9 Mediated Gene Editing in Long-Term Engrafting Human Hematopoietic Stem/Progenitor Cells. Blood 128, 2312–2312 (2016).]

As noted in Supplementary Table 8, our mean 1.8% absolute HDR levels for short base pair changes are comparable and better in some cases, than those reported in literature to date for comparable cell types. With respect to absolute levels of editing, there is notable variation across cell types and loci, as indicated in Supplementary Table 8 and noted previously in literature [e.g., Miyaoka, Y. et al. Scientific Reports 6, srep23549 (2016)]: therefore, we have focused on the stronger claim of shifting the balance of HDR:NHEJ outcomes. This shift in HDR:NHEJ balance is the central finding of this study and utility of the S1mplex tool. The importance and potential impact of this finding is contextualized by recent papers:

In hPSCs, a recent study explicitly stated the challenge regarding the ratio of NHEJ to HDR:

"A major challenge for precise genome-editing is the ability to induce high-fidelity HDR edits with a low NHEJ background. For example, in our attempts to isolate human induced pluripotent stem cell (iPSC) lines with genomic modifications via HDR, multiple isolated iPSC lines had one allele with desirable HDR and disruption of the other allele by NHEJ...These observations highlight the importance of minimizing the NHEJ activity to achieve precise genome-editing." from Miyaoka, Y. et al. Systematic quantification of HDR and NHEJ reveals effects of locus, nuclease, and cell type on genome-editing. Scientific Reports 6, srep23549 (2016).

In mice, a practical guide to gene editing highlighted a related key problem with NHEJ and the impact that increased HDR could have:

"A complicating issue with the CRISPR system is actually the robustness of Cas9. The majority of gRNA-targeted Cas9-induced DSBs are eventually channeled to an error prone NHEJ repair pathway...this robustness becomes problematic when the goal is to induce precise mutations into a locus via HDR with an introduced template. If a DSB is first repaired by NHEJ in a manner that precludes subsequent Cas9:gRNA recognition or cutting (for example by mutating the PAM site), then the desired modification is thwarted." from Singh, P., Schimenti, J. C. & Bolcun-Filas, E. A Mouse Geneticist's Practical Guide to CRISPR Applications. Genetics 199, 1–15 (2015).

This year, leaders in this field, Jennifer Doudna and Jacob Corn, along with colleagues acknowledge that absolute levels of editing can vary across cell types and summarize the impact that higher HDR could have for many genetic diseases: “...although CRISPR–Cas knockouts are effective [predominately by NHEJ] in nearly any cell, rates of HDR can vary across cell types...These barriers are particularly frustrating, because sequence insertion or replacement in these contexts could be used to model or to treat many genetic diseases.” From Fellmann, C., Gowen, B. G., Lin, P.-C., Doudna, J. A. & Corn, J. E. Cornerstones of CRISPR-Cas in drug discovery and therapy. *Nat Rev Drug Discov* **16**, 89–100 (2017).

A wide-ranging assessment of the field in *Science* last year also explicitly mentioned that this issue as bottleneck in the development of CRISPR therapeutics: “Using CRISPR to cut out part of a gene—not correct the sequence—is relatively easy to do...But the low rate of HDR in most cells is one reason why the first use of CRISPR in the clinic will likely involve disrupting genes, not fixing them.” from Kaiser, J., The gene editor CRISPR won’t fully fix sick people anytime soon. Here’s why. *Science* (2016).

The minimization of NHEJ could bypass these bottlenecks in gene editing and enable serial editing strategies to further increase the absolute levels of HDR. As stated in the Discussion, our S1mplex tool is an important addition to the toolkit of genome editors, as it is powerful and complementary to other work in the field: the S1mplex strategy could be used with small molecules that perturb HDR and NHEJ pathways and sorting strategies to enrich for precisely-edited cells. We demonstrate sorting with our S1mplex strategy to isolate HDR edited clones at an efficiency of 24% in Figure 6b.

4) There are a number of apparent inconsistencies and peculiarities in the manuscript, of which I note only a few:

The diagrams in Figure 1b do not depict sgRNA structure accurately. The natural stem loops should be shown explicitly, including the cases in which they are interrupted. Both the SL2 and SL3 RNAs have long stems that come neither from standard sgRNA nor the S1m aptamer. This should be acknowledged. Finally, the insertion in the SL3 construct is not in stem-loop 3, but follows that feature and is really an addition to the 3’ end.

We agree with the reviewer that the structure and sgRNA modifications could be illustrated and named more accurately. We have now amended Figure 1b to indicate predicted base pairing within the sgRNA:

This image is consistent with illustrations of modified sgRNA in Ma et. al Nature Biotechnology, 2016, and should provide the reader with a more accurate illustration of the structure of the modified sgRNAs used in this study. Further, we added discussion in the Main Text regarding use of a longer stem RNA sequence than what was originally used in sgRNA-aptamer work (Shechner et al. Nature Methods, 2015).

5) Figure 2a indicates that addition of streptavidin shifts all of the Cas9-S1m-sgRNA complex, while multiple other experiments show that the binding of streptavidin to this RNA is quite weak (Supp. Figs. 1b, 3b, 3c).

We do not agree with the reviewer regarding the strength of this binding. Published data on the S1m aptamer indicates a dissociation constant (K_d) of 29 nM (Leppek and Stoecklin, Nucleic Acids Research, 2014) which is similar to the affinity of many antibodies. The gel data in Supplementary Figures 1b, 3b, 3c are informative in a semi-quantitative manner to distinguish binding partners. In the revision, we added provided immunocytochemistry co-localization data within cells and DLS data in Figure 2a that may be used in a more quantitative fashion. We also note that the sgRNA alone is below the limit of detection of the of dynamic light scattering (DLS) instrument, because the principles of DLS used for traces shown in Figure 2a does not sense

nucleic acids well. For example, below is a DLS trace from sgRNA alone, now shown in Supplementary Figure 1c:

c. DLS trace of free sgRNA alone in solution. Nucleic acids are not robustly detected by this DLS instrument.

Therefore, the DLS traces in Figure 2a in the presence of 2.5X molar excess sgRNA will not show any unbound sgRNA.

The biotin challenge experiment in Supplemental Figure 3c with 4X molar excess of biotin relative to S1m-sgRNA indicates that the binding of streptavidin is stable with the stoichiometry used in this study, and even in extreme conditions.

c. Competition of biotin with S1m-sgRNA-SL1 after binding to streptavidin protein. 4-fold excess biotin was added to S1m-sgRNA-SL1-streptavidin complexes and incubated for 0, 5, 10, 20, and 30 minutes. No significant change was seen even after 30 minutes of competition.

Further, the co-localization data of streptavidin with Cas9 in Figure 2 within cells indicates that the S1m-streptavidin binding persists throughout delivery and subcellular trafficking into the nucleus. Together, these data indicate that the binding of streptavidin within the S1mplex is strong.

6) Several controls are missing from Supp. Fig. 3b that are necessary to identify individual bands: S1m-sgRNA alone; S1m-sgRNA + streptavidin, but without Biotin-ssODN; Biotin-ssODN + streptavidin (without sgRNA).

We have performed the gel with the suggested additional controls and replaced Supplementary Figure 3b with the image below. The full gel image is on the next page.

Free ssODN runs closer to the S1mplex formation in this gel due to the use of longer ssODN than in previous experiments used in the revision. The positions of these control conditions described by the reviewer in lanes 3, 7, and 15 are as expected. The conclusions from this experiment still hold as described in the manuscript.

7) I have been reliably informed that the Amnis ImageStreamX Mark II Imaging Flow Cytometer does not produce phase contrast images, so it is puzzling where the pictures in Figure 2b come from.

We thank the reviewer for noting this issue. The original figure should have labeled this panel as "Brightfield". We have updated Figure 2b accordingly, as shown below.

In the previously revised manuscript, the related Supplementary Figure 2 had the correct brightfield (Ch1) label in legend.